# LEGO: An LLM-Enabled Hierarchical Optimizer for Tensor Computation Graphs with Structure-Aware Search and Compositional Synthesis

Ruiyuan Xu [* 1 2]  Shuoming Zhang [* 1 2]  Guangli Li [1 2 3]  Qiuchu Yu [1 2]  Rui Zhang [1]  Yangyu Zhang [1 2]  Hao Qian [3]
Chunwei Xia [4]  Jiacheng Zhao [1 2]  Chenxi Wang [1 2]  Xiaobing Feng [1 2]  Jingling Xue [3]  Huimin Cui [1 2]

## Abstract

Automating end-to-end GPU kernel generation with Large Language Models (LLMs) faces a critical tension between *global performance* and *exploration efficiency*. We present LEGO, a hierarchical framework that resolves this trade-off via a parallel multi-agent search over a recursive AND-OR FusionTree. LEGO synergizes two complementary flows: *Top-Down Construction* decomposes complex graphs into valid, context-isolated sub-problems to guarantee correctness and enable parallel exploration, while *Bottom-Up Mutation* speculatively fuses verified sub-plans to recover global locality for peak performance. This bi-directional mechanism effectively prunes the search space to avoid repetitive unguided sampling, while naturally parallelizing exploration, and enabling the discovery of sophisticated fusion strategies. Evaluations demonstrate that LEGO achieves **2.18×–13.48×** speedups over PyTorch Eager and, compared to monolithic baselines, reduces end-to-end exploration time by up to **2.47×** and token consumption by up to **7×**, across diverse end-to-end models. Code is available at https://github.com/mmt-at/LEGO.

## 1. Introduction

Large Language Models (LLMs) have recently exhibited remarkable capabilities in synthesizing high-performance tensor programs (Hammond et al., 2025; Wang & PyTorch Team at Meta, 2025; Dong et al., 2025; Woo et al., 2025; Li

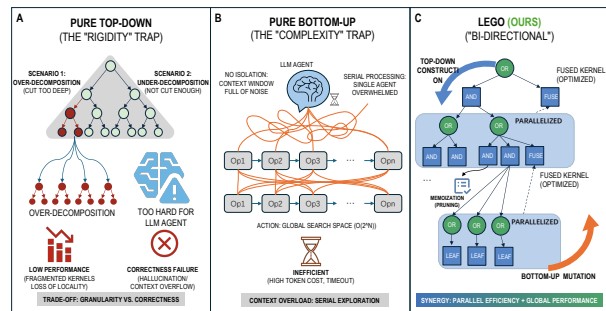

*Figure 1.* **Paradigms of LLM-based Kernel Synthesis. (A)** **Pure Top-Down** faces a "Rigidity Trap": struggling between fragmentation (performance loss) and context overflow (hallucination). **(B)** **Pure Bottom-Up** faces a "Complexity Trap": the exponential ($O(2^N)$) global search space causes contextual overload and inefficient serial exploration. **(C)** **LEGO (Ours)** resolves this via a **Bi-Directional FusionTree**, synergizing structure-aware **parallel construction** (correctness) with **speculative fusion** (performance).

et al., 2025; Su et al., 2025), often rivaling hand-crafted kernels and expert-tuned compiler outputs. These advances suggest a new paradigm for deep learning system optimization, in which code generation is not limited to predefined lowering rules but can be dynamically synthesized. However, translating this local synthesis capability into end-to-end model-level optimization remains non-trivial. Optimizing tensor computation graphs necessitates coordinating graph partitioning, operator fusion, and implementation selection across highly interdependent design choices, forming a combinatorial search problem that extends far beyond isolated kernel generation.

Traditional tensor compilers, such as TorchInductor (Ansel et al., 2024) and XLA (OpenXLA, 2026), address this challenge through heuristic-driven graph rewriting and predefined fusion patterns, which provide efficient but structurally inflexible optimization strategies. Search-based approaches, such as TASO (Jia et al., 2019), enlarge the exploration space but often struggle with the exponential complexity of graph transformation.

Existing approaches typically tackle this complexity through unidirectional strategies, yet both extremes fall short when applied to LLMs. As depicted in Figure 1, **Pure bottom-up**

*Equal contribution [1] Institute of Computing Technology, Chinese Academy of Sciences, Beijing, China [2] University of Chinese Academy of Sciences, Beijing, China [3] School of Computer Science and Engineering, University of New South Wales, Sydney, Australia [4] School of Computer Science, University of Leeds, Leeds, United Kingdom. Correspondence to: Guangli Li <liguangli@ict.ac.cn>.

*Proceedings of the 43rd International Conference on Machine Learning*, Seoul, South Korea. PMLR 306, 2026. Copyright 2026 by the author(s).

**(or flat) approaches**, which scale agentic kernel generation (Hammond et al., 2025) to an entire flattened graph, expose maximal fusion opportunities but suffer from contextual overload, leading to hallucinations and prohibitive sampling costs due to the exponential search space. Conversely, **pure top-down decomposition** (Wang & PyTorch Team at Meta, 2025) ensures manageable context and high generation success rates (correctness) but often fragments critical computation boundaries, sacrificing valuable cross-operator fusion opportunities (performance). We argue that achieving both scalable exploration and peak performance requires a *bi-directional synergy*: utilizing structural decomposition to guarantee validity, while retaining the ability to recover global locality through fusion.

To realize this, we utilize the inherent structural regularities of modern neural networks—composed of repeated blocks and nested motifs—as a strong prior. We formulate the optimization process as a search over a recursive **AND-OR FusionTree**. Unlike rigid partitioning, this hierarchical structure supports a flexible Construct-then-Mutate paradigm. Crucially, the FusionTree naturally isolates subproblems, allowing **parallel multi-agent exploration** along the tree structure. This decouples the exploration time from the model size, limiting the cost to the critical path of the decomposition.

We present LEGO, a hierarchical LLM-driven optimizer that integrates top-down structure-aware decomposition with bottom-up compositional synthesis for scalable tensor graph optimization. LEGO aligns optimization boundaries with the inherent structural regions of neural network models, performing structure-guided decomposition to generate well-formed subproblems. Within each region, LEGO explores alternative realizations through compositional synthesis, including direct region-level code generation and structured fusion across subregions. By coupling structural decomposition with systematic evaluation of competing implementations, LEGO enables continuous refinement of graph partitions, allowing optimized subregions to inform higher-level composition decisions. Evaluation on representative workloads, including ResNet, Qwen, Stable Diffusion, Mamba, and mHC, shows that LEGO achieves $2.18\times-13.48\times$ speedups over PyTorch Eager, consistently improving end-to-end performance over both heuristic-based tensor compilers and existing LLM-driven optimization approaches.

This paper makes the following contributions:

1. **Structure-Aware Search.** We formulate LLM-based tensor graph optimization as a structure-aware recursive decision problem, revealing that the hierarchical regularities of models induce an AND-OR search space over graph decompositions and implementations.

2. **Construct-then-Mutate Paradigm.** We present LEGO, a hierarchical optimizer that integrates top-down decomposition (Construction) with bottom-up speculative fusion (Mutation). This synergy allows the system to efficiently navigate the trade-off between modular scalability and cross-boundary optimization.

3. **SOTA Performance with High Exploration Efficiency.** Evaluation demonstrates that LEGO achieves $2.18\times-13.48\times$ speedups over PyTorch Eager across diverse architectures (CNNs, LLMs, SSMs). Critically, LEGO achieves these gains with up to $7\times$ lower token consumption and $2.47\times$ faster search time compared to flat LLM baselines, establishing a new Pareto frontier for cost-effective kernel synthesis.

## 2. Related Work

**Tensor Compilers and Auto-Tuners.** Mainstream AI compilers like XLA (OpenXLA, 2026) and TorchInductor (Ansel et al., 2024) rely on lowering passes and predefined heuristics. While efficient in compilation time, they often miss complex fusion opportunities. Search-based auto-tuners, such as Ansor (Zheng et al., 2020) and AutoTVM (Chen et al., 2018), explore schedule spaces but are prohibitively expensive and constrained by heuristic-based graph partitioning. Recent tile-level DSL compilers such as TileLang (Wang et al., 2026; Cheng et al., 2025) and super-optimizers such as Mirage (Wu et al., 2025) achieve strong per-kernel performance but require manual identification of target regions and do not provide end-to-end automation. Consequently, a gap remains between per-operator compiler optimization and full-graph model-level performance.

**LLM-Driven Kernel Synthesis.** Recent work has shown that LLMs can synthesize high-performance Triton/CUDA kernels for single operators and small subgraphs, often rivaling compiler-generated code (Hammond et al., 2025; Li et al., 2025; Su et al., 2025; Woo et al., 2025; Dong et al., 2025; Wang et al., 2025). These systems typically rely on iterative code generation with execution-based feedback (compile errors, numerical mismatches, or micro-benchmark regressions) to steer the model toward correct and fast kernels. However, scaling from Level-1 operators to Level-3 end-to-end models remains challenging due to context limits, long-range dependencies, and fragile global correctness constraints.

**Agentic Systems for Subgraph and End-to-End Synthesis.** Recent work increasingly adopts multi-agent workflows that decompose kernel generation into planning, coding, and debugging/tuning roles (Dong et al., 2025; Du et al., 2025; Wei et al., 2025; Zhang et al., 2025; Sereda et al., 2025). Some further scale via manager–worker delegation or reflection-based refinement for specific back-

ends and vendors (Wang & PyTorch Team at Meta, 2025; Wang et al., 2025). LEGO distinguishes itself through its structural optimization prior and recursive refinement paradigm. Instead of treating graph partitioning as a one-shot preprocessing step, LEGO performs iterative, region-based refinement: it progressively decomposes the graph into hierarchical regions, and feeds validated subregion results back to guide higher-level AND–OR decisions over `split/fuse/codegen`. Moreover, LEGO serves as a backend-agnostic meta-optimizer that can orchestrate existing compilation backends rather than replace them.

**Structure-Awareness and Scalability.** Structure-aware optimization has long been a key strategy for scaling graph search and compilation. Search-based graph optimization systems such as TASO (Jia et al., 2019) explore graph substitutions, while DyPARS (Qian et al., 2026) highlights the effectiveness of structure-aware search for dynamic shapes. LEGO builds on this line by using `nn.Module` boundaries as a practical structural prior for safe divide-and-conquer, and by coupling structure-aware decomposition with LLM-enabled local synthesis and end-to-end recomposition.

# 3. Methodology

We present LEGO, a hierarchical compilation framework designed to bridge the gap between high-level structural abstraction and low-level kernel optimization (Figure 2). LEGO formulates the optimization process as a search problem over a recursive AND-OR tree, termed the *FusionTree*. In this section, we first articulate the design philosophy, followed by the formal definition of the search space and the **baseline-guided recursive search algorithm**.

## 3.1. Design Philosophy

Current deep learning compilers (e.g., TorchInductor (Ansel et al., 2024), XLA (OpenXLA, 2026)) typically flatten the model into a monolithic computational graph to expose global fusion opportunities. While theoretically enabling arbitrary operator fusion, this approach relies on a *flat trace analysis* that scales linearly with the number of operators ($O(|\mathcal{V}|)$). Crucially, this flattening process obscures high-level architectural intent, treating semantically identical structures (e.g., distinct Attention heads) as unrelated subgraphs. LEGO challenges this paradigm based on two key insights:

**Pruning via Structural Isomorphism.** Deep neural networks exhibit pervasive structural regularity. From ResNet blocks (He et al., 2016) to Transformer layers (Brown et al., 2020), models are composed of massive sequences of isomorphic modules. Unlike existing compilers where minor context variations in a flattened trace can lead to inconsistent decisions for identical blocks, LEGO leverages **source-level**

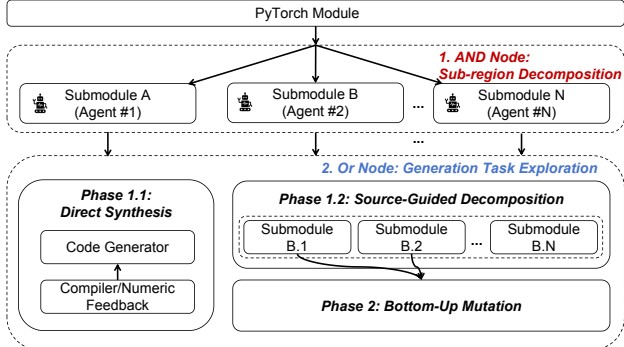

*Figure 2.* **Overview of the LEGO Framework.** A PyTorch module is recursively decomposed into sub-regions dispatched to parallel agents (AND-node decomposition). Each sub-region is solved as an OR-node that explores **top-down construction**— *Phase 1.1* direct kernel synthesis with compiler/numerical feedback and *Phase 1.2* source-guided decomposition—followed by *Phase 2* **bottom-up mutation**, which speculatively fuses verified sub-plans. The minimal-latency plan is retained and propagated upward.

**isomorphism** as a strong prior.

**Structured Decomposition with Speculative Relaxation.** A strictly hierarchical approach, while scalable, risks severing critical fusion opportunities that span across module boundaries (e.g., fusing the output of a Residual Block with the input of the next LayerNorm, or fusing the last decoder block with the output-layer SoftMax). To resolve the tension between "modular scalability" and "global optimality", LEGO adopts a **Construct-then-Mutate** paradigm. We utilize the source-code hierachy as a robust skeleton for top-down decomposition (*Construction*), ensuring search tractability. Simultaneously, we treat these structural boundaries not as rigid walls, but as soft constraints that can be "short-circuited" by the LLM. Through bottom-up *Mutation*, the system speculatively identifies and heals inefficient boundaries, effectively combining the safety of structural compilation with the performance ceiling of flat optimization.

## 3.2. Problem Formulation: The Recursive FusionTree

We define the kernel generation task as a recursive cost-minimization problem. Let $\mathcal{G}$ denote the computational graph of a model and $\mathbb{S}$ be the infinite space of all valid semantic-preserving implementations. For any region $\mathcal{R} \subseteq \mathcal{G}$, our theoretical objective is to find the global optimum:

$$V^*(\mathcal{R}) = \min_{\pi \in \mathbb{S}(\mathcal{R})} \text{Latency}(\pi) \tag{1}$$

Ideally, this recursive optimization satisfies the Bellman

optimality principle:

$$V^*(\mathcal{R}) = \min \left\{ \min_{\pi \in \mathbb{A}_{\mathrm{at}}} C(\pi), \ \min_{d \in \mathbb{D}_{\mathrm{all}}} \left( \epsilon + \sum_{r \in d(\mathcal{R})} V^*(r) \right) \right\} \tag{2}$$

where $\mathbb{A}_{\mathrm{at}}$ represents all possible atomic (monolithic) kernel implementations (e.g., via manual CUDA writing), $\mathbb{D}_{\mathrm{all}}$ represents all valid topological partitions of the graph, and $\epsilon$ denotes the kernel launch overhead.

**Approximation via Heuristic Action Space.** Directly solving Eq. 2 is intractable due to the combinatorial explosion of the partitioning space $\mathbb{D}_{\mathrm{all}}$. Standard compilers typically approximate this by performing greedy fusion on flattened traces. However, as argued in subsection 3.1, this "flatten-then-rebuild" approach discards valuable semantic information.

LEGO adopts a different approximation strategy. We contend that the **Source-Code Hierarchy** serves as a strong topological prior, effectively reducing the vast space $\mathbb{D}_{\mathrm{all}}$ to a tractable subspace $\mathbb{D}_{\mathrm{source}}$. To recover fusion opportunities that might be severed by strict source boundaries, we construct the action space $\mathcal{A} = \mathcal{A}_{\mathrm{const}} \cup \mathcal{A}_{\mathrm{mut}}$ as an interplay between top-down decomposition and bottom-up relaxation:

- **Construction Space ($\mathcal{A}_{\mathrm{const}}$):** This set defines the **Top-Down** exploration. Instead of arbitrary graph cuts, we restrict the decomposition operator to align with source-level modularity:

$$\mathcal{A}_{\mathrm{const}} = \{ \mathrm{Base}(\mathcal{R}), \mathrm{Gen}(\mathcal{R}), \mathrm{Split}_\Phi(\mathcal{R}) \} \tag{3}$$

  where Base and Gen represent atomic implementations (Library/LLM), and $\mathrm{Split}_\Phi$ decomposes $\mathcal{R}$ strictly along the boundaries defined by the isomorphism signature $\Phi$ (i.e., `nn.Module` borders). This heavily prunes the search tree by leveraging the developer's architectural intent.

- **Mutation Space ($\mathcal{A}_{\mathrm{mut}}$):** Solely relying on source boundaries may miss optimizations that span across modules. To recover these lost opportunities, we define $\mathcal{A}_{\mathrm{mut}}$ as a **Bottom-Up** relaxation. We model mutation as an **LLM-parameterized sampling process** over the verified sub-plans $\Pi$:

$$\mathcal{A}_{\mathrm{mut}} = \{ \mathrm{Fuse}_{\mathrm{all}}(\Pi) \} \cup \{ \mathrm{Fuse}_{\mathrm{sub}}^{(i)}(\Pi) \}_{i=1}^K \tag{4}$$

  Here, $\mathrm{Fuse}_{\mathrm{sub}}^{(i)} \sim P_{\mathrm{LLM}}(\cdot | \Pi)$ represents a probabilistic attempt to "short-circuit" the structural split. The LLM analyzes the data dependencies of $\Pi$ to propose a fused subset (e.g., a diamond subgraph) that transcends the original split boundaries. In our implementation, we set the sampling budget $K = 1$ for efficiency.

Consequently, we solve the *Surrogate Bellman Equation*:

$$V(\mathcal{R}) \approx \min \left\{ \underbrace{\min_{f \in \mathcal{A}_{\mathrm{const}}} \mathrm{Eval}(f(\mathcal{R}))}_{\text{Phase I: Construction}}, \ \underbrace{\min_{g \in \mathcal{A}_{\mathrm{mut}}} \mathrm{Eval}(g(\mathcal{D}^*(\mathcal{R})))}_{\text{Phase II: Mutation}} \right\} \tag{5}$$

Here, $\mathcal{D}^*(\mathcal{R})$ denotes the verified sub-plans $\Pi$ obtained from the best source-guided decomposition found in Phase I.

The **FusionTree** realizes this approximation through two alternating node types:

- **OR Nodes ($v_{\mathcal{R}}$):** Represent an optimization goal for region $\mathcal{R}$. Instead of a flat search, the agent executes a *Phased Exploration Protocol* that interleaves top-down construction with bottom-up speculation:

  **Phase I: Top-Down Construction.** The agent first attempts to solve the region atomically or decompose it recursively:

  0. **Initialization (Baseline):** Before initiating optimization, the agent benchmarks the raw module using standard backends (PyTorch Eager, `torch.compile`) and, if applicable, vendor-optimized libraries. The minimum latency among these candidates establishes the **Verified Upper Bound** ($t_{base}$). All subsequent strategies are considered valid only if they strictly outperform this baseline.
  1. **Direct Synthesis (Codegen):** The agent prompts the LLM to generate a specialized Triton kernel for $\mathcal{R}$. This candidate undergoes a rigorous validation pipeline—encompassing functional correctness, numerical precision, and serialized performance profiling (`measure`, detailed in Algorithm 4). This step aims to surpass the baseline via bespoke kernel logic ($t_{gen}$).
  2. **Source-Guided Decomposition (Split):** The agent applies source-guided partitioning (e.g., leveraging `nn.Module` boundaries) to decompose $\mathcal{R}$ into sub-components. This yields a "divide-and-conquer" solution ($t_{split}$), validated by ensuring the recombined graph signature matches the original isomorphism hash $\Phi(\mathcal{R})$.

  **Phase II: Bottom-Up Mutation.** *Conditioned on the success of Step 2*, if decomposition yields a set of valid child nodes, the agent attempts to "mutate" the resulting structure to discover superior fusion configurations:

  3. **Aggressive Full-Fusion:** The agent attempts to fuse all verified sub-components from Step 2 back into a single monolithic kernel. This bottom-up reduction targets maximum data locality ($t_{full}$), offering a competitive alternative to the split strategy, albeit with higher resource pressure.

4. **Selective Partial-Fusion:** Using the dependency graph of the valid children, the agent explores a *hybrid reduction*. The LLM analyzes the topology to autonomously propose a partial fusion plan—clustering high-affinity operators (e.g., a computation-heavy diamond subgraph) while retaining the structural cut for others. This mutation ($t_{part}$) balances kernel complexity with structural flexibility.

Finally, the OR node acts as a *selector*, finalizing its state by retaining only the optimal plan with the minimal verified latency to propagate upward.

- **AND Nodes** ($u_d$)**:** Represent a specific *strategy* derived from the parent OR node. These nodes fall into two categories:

  - *Terminal AND:* A concrete implementation (e.g., a specific Triton kernel source code or a vendor library call).
  - *Structural AND:* A decomposition plan that splits $\mathcal{R}$ into disjoint sub-regions $\{\mathcal{R}_1, \ldots, \mathcal{R}_k\}$.

Since the execution of these sub-regions is serialized by data dependencies, the cost of a structural AND node corresponds to the aggregate latency of its children plus inevitable kernel launch overheads.

**Global Memoization.** To exploit structural redundancy, we define a signature function $\Phi : \mathcal{R} \rightarrow \mathbb{S}$ encompassing operator topology, tensor shapes, and data types. We maintain a global memoization table $\mathcal{M}$. Before expanding any OR-node $v_{\mathcal{R}}$, we check if $\Phi(\mathcal{R}) \in \mathcal{M}$. A hit allows immediate pruning, retrieving the optimal plan $\pi^*$ computed for an isomorphic region.

### 3.3. Algorithm: Dynamic Tree Construction

We formalize the core logic in Algorithm 1. Note that for brevity, we omit the global memoization checks which are implicitly performed at the beginning of each function call.

The algorithm follows a hierarchical **Baseline-Guided Expansion** paradigm: it first anchors the search with a verified baseline (Step 1) and then spawns parallel agents to find superior strategies via decomposition (Step 2) and speculative fusion (Step 3 & 4).

To illustrate the end-to-end workflow, Figure 3 depicts the optimization trajectory of the Manifold-Constrained Hyper-Connections (mHC) workload. The **Top-Down Construction** agents first analyze the AST to identify three semantically distinct stages: (1) pre-mapping GEMM operations, (2) the iterative Sinkhorn projection, and (3) post-mapping fusion. Each stage is then independently dispatched for code generation. Notably, the Sinkhorn iteration is recognized as

---

**Algorithm 1** Recursive FusionTree Construction

1: **Function** BuildFusionTree($\mathcal{R}$):
2: **Input:** Region $\mathcal{R}$
3: **Output:** Optimal Plan $\pi^*$, Latency $t^*$
4: *// Phase I — Top-Down Construction: Establish Baseline (Step 0)*
5: *// Run via Eager/TorchCompile to get verified upper bound*
6: $t_{base}, \pi_{base} \leftarrow$ Measure(Eager $\cup$ TorchCompile($\mathcal{R}$))
7: *// Phase I — Top-Down Construction: Atomic Codegen (Step 1)*
8: $t_{gen}, \pi_{gen} \leftarrow$ Measure(LLM_Codegen($\mathcal{R}$))
9: *// Update local best (keep the lower-latency plan)*
10: $(t_{curr}, \pi_{curr}) \leftarrow \min\{(t_{base}, \pi_{base}), (t_{gen}, \pi_{gen})\}$
11: *// Phase I — Top-Down Construction: Recursive Decomposition (Step 2)*
12: $\{\mathcal{R}_1, \ldots, \mathcal{R}_k\} \leftarrow$ SourceGuidedSplit($\mathcal{R}$)
13: *// Solve sub-problems in parallel*
14: **for all** $\mathcal{R}_i$ **in parallel do**
15:    $task_i \leftarrow$ **spawn** BuildFusionTree($\mathcal{R}_i$)
16: **end for**
17: **sync** $\{task_1, \ldots, task_k\}$
18: $t_{split} \leftarrow \epsilon + \sum_i t^*_{child\_i}$    *// add kernel launch overhead $\epsilon$ (AND-node rule)*
19: $\pi_{split} \leftarrow \{\pi^*_{child\_i}\}$    *// structural-AND plan*
20: *// Phase II — Bottom-Up Mutation: Speculative Fusion (Steps 3 & 4)*
21: $Leaves \leftarrow$ CollectLeaves($\{task_i.\pi^*\}$)
22: $Candidates \leftarrow$ SpeculateFusion($Leaves$)
23: $t_{fuse}, \pi_{fuse} \leftarrow \min_{s \in Candidates}$ Measure($s$)
24: *// Reduction: Select Best Plan*
25: *// Select the strategy with minimal latency*
26: $S \leftarrow \{(t_{curr}, \pi_{curr}), (t_{split}, \pi_{split}), (t_{fuse}, \pi_{fuse})\}$
27: $t^*, \pi^* \leftarrow$ SelectBest($S$)
28: **return** $\pi^*, t^*$

---

a loop-intensive hotspot, prompting the agent to synthesize a specialized Triton kernel rather than a standard PyTorch sequence.

### 3.4. Optimization Dynamics: Latency Propagation

Unlike heuristic search algorithms (e.g., A*) that rely on estimated lower bounds, LEGO employs a **Baseline-Guided Recursive Search**. The optimization process propagates verified latency values bottom-up to identify the global optimum defined by the Surrogate Equation (Equation 5).

**Latency Propagation Rules.** The cost of any node in the FusionTree is derived strictly from verified execution measurements. The propagation follows two rules:

- **Summation Rule (AND Nodes):** For a structural

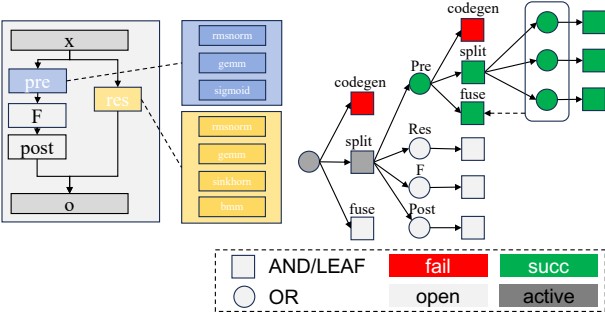

*Figure 3.* Case study of Manifold-Constrained Hyper-Connections (mHC).

AND-node $u_d$ decomposing $\mathcal{R}$ into sub-regions $\{\mathcal{R}_1, \ldots, \mathcal{R}_k\}$, the total latency accounts for both computation and system overheads ($\epsilon$):

$$V(u_d) = \epsilon + \sum_{i=1}^{k} V(v_{\mathcal{R}_i}) \qquad (6)$$

- **Minimization Rule (OR Nodes):** For an OR-node $v_{\mathcal{R}}$, the latency is determined by the competitive race among the strategies defined in the *Phased Exploration Protocol*:

$$V(v_{\mathcal{R}}) = \min\left(t_{base}, t_{gen}, t_{split}, t_{full}, t_{part}\right) \quad (7)$$

## 4. Evaluation

**Workloads.** We evaluate LEGO across a diverse suite of five representative workloads spanning distinct architectural paradigms: **ResNet** (He et al., 2016) (CNN), **Qwen3** (Yang et al., 2025) (LLM), **SD3-MMDiT** (Esser et al., 2024) (Diffusion Transformer), **Mamba2** (Dao & Gu, 2024) (State-Space Model), and **mHC-MoE** (Xie et al., 2025) (Emerging Architecture).

**Environment.** To demonstrate hardware generalization, we perform experiments using a single-GPU configuration across two generations of NVIDIA architectures: the workstation-grade RTX 6000 Ada and the datacenter-grade H100 (Hopper). The software environment is standardized on CUDA 12.9.0, cuDNN 9.10.2, PyTorch 2.9.1, and Triton 3.5.1. Unless otherwise specified, we report the median end-to-end latency of the model forward pass, measured over repeated runs after a warm-up period. For the agent backend model, we utilize **GPT-5** (OpenAI, 2026) (temperature=1.0) to drive hierarchical exploration.

### 4.1. Baselines and Metrics

**Baselines.** We compare LEGO against standard execution tiers and existing LLM-based agents (we additionally compare against the TileLang DSL compiler in Appendix D):

- **PyTorch Eager**: Standard interpreter-based execution, the universal reference ($1.0\times$).
- **Inductor (Default/Max-Autotune)**: `torch.compile` with default or aggressive (`max-autotune`) optimizations.
- **LLM Agents**: KernelAgent (KernelFalcon) (Wang & PyTorch Team at Meta, 2025), CudaForge (Zhang et al., 2025), and AKG (Du et al., 2025)—representative open-source PyTorch-to-kernel agents under matched GPT-5 budgets.
- **Monolithic Baseline**: An iterative LLM agent (similar to (Hammond et al., 2025)) that optimizes the flattened graph without structural decomposition, serving as the reference for exploration cost.
- **Ours (LEGO)**: LEGO-synthesized kernels with system-level optimizations (e.g., CUDA Graphs).

**Metrics.** We report metrics across three dimensions, with baselines adjusted for the specific experimental context:

**Inference Performance:**

- **End-to-End Latency** ($L$): The median wall-clock time (ms) of the model forward pass, measured over repeated runs after warm-up.
- **Inference Speedup** ($S$): Calculated as $S = L_{\text{Base}}/L_{\text{LEGO}}$. We employ two distinct baselines depending on the evaluation goal:
  - **vs. Eager:** Used in the main performance benchmarks (Table 1) to quantify total system acceleration.
  - **vs. Best Inductor:** Used in ablation studies (Table 2) to rigorously quantify the marginal gain of LEGO over the strongest available compiler optimizations, in most cases, it's Max-Autotune.

**Search Efficiency (vs. Monolithic Baseline):**

- **Exploration Time Speedup:** The reduction in wall-clock time required to discover a valid solution.
- **Token Reduction Factor:** The ratio of total tokens consumed (prompt + completion) by the monolithic baseline versus LEGO.

**Correctness:** We enforce strict numerical correctness. A generated kernel is considered valid only if its output matches the reference (Eager) within a tolerance of `atol=1e-2`, `rtol=1e-2`, aligned with most related work (Ouyang et al., 2025; Dong et al., 2025; Wang & PyTorch Team at Meta, 2025; Li et al., 2025).

### 4.2. Evaluation Harness

For microbenchmarks and ablations, we use a unified benchmarking harness to compare a reference task module against an optimized (or candidate) module under identical inputs

*Table 1.* **End-to-end speedup over PyTorch Eager.** LLM agents use GPT-5 with matched budgets (pass@10). Max-AT: `max-autotune`; **Bold**: best; underline: best baseline; $\times$: generation failure;

| Model | Eager (ms) | Inductor | | LLM Agents | | | **LEGO** |
| | | Default | Max-AT | KernelAgent | CudaForge | AKG | |
|---|---|---|---|---|---|---|---|
| | | | | NVIDIA RTX 6000 ADA | | | |
| Mamba2-1.3B | 1416.85 | 1.77$\times$ | 1.84$\times$ | 0.74$\times$ | $\times$ | $\times$ | **2.18$\times$** |
| SD3-MMDiT | 1999.98 | 1.62$\times$ | 1.84$\times$ | 0.99$\times$ | $\times$ | $\times$ | **4.09$\times$** |
| mHC-MoE | 19.50 | 0.82$\times$ | 0.91$\times$ | $\times$ | $\times$ | 0.44$\times$ | **13.48$\times$** |
| ResNet18 | 2.40 | 1.76$\times$ | 5.33$\times$ | 0.17$\times$ | 1.75$\times$ | 0.93$\times$ | **6.00$\times$** |
| Qwen3-8B | 1264.18 | 1.31$\times$ | 1.63$\times$ | $\times$ | $\times$ | $\times$ | **2.21$\times$** |
| | | | | NVIDIA H100 | | | |
| Mamba2-1.3B | 810.93 | 2.79$\times$ | 2.82$\times$ | $\times$ | $\times$ | $\times$ | **3.68$\times$** |
| SD3-MMDiT | 1100.17 | 2.11$\times$ | 2.14$\times$ | $\times$ | $\times$ | $\times$ | **6.13$\times$** |
| mHC-MoE | 13.73 | 0.75$\times$ | 0.82$\times$ | 0.99$\times$ | $\times$ | 0.99$\times$ | **9.26$\times$** |
| ResNet18 | 1.77 | 1.46$\times$ | 3.87$\times$ | 0.90$\times$ | 1.42$\times$ | 0.98$\times$ | **3.96$\times$** |
| Qwen3-8B | 552.41 | 1.92$\times$ | 1.95$\times$ | $\times$ | $\times$ | $\times$ | **2.84$\times$** |

and execution settings. To ensure fair comparisons, we optionally apply a *state_dict mapping spec* so that the reference and candidate share the same managed weights without importing Python modules with side effects. We collect latency (ms) over repeated runs under three modes (*eager*, *default*, *max-autotune*); in practice, we report median latency.

### 4.3. End-to-End Results

Table 1 summarizes end-to-end performance. LEGO achieves 2.18$\times$–13.48$\times$ speedups on RTX 6000 Ada and 2.84$\times$–9.26$\times$ on H100 over PyTorch Eager, consistently outperforming TorchInductor across all workloads. The largest gains occur on emerging operators (mHC-MoE, SD3-MMDiT) where rigid compiler heuristics miss profitable fusion patterns, demonstrating LEGO's "zero-day" optimization capability for novel architectures. Table 1 also includes three representative open-source kernel-generation agents (Wang & PyTorch Team at Meta, 2025; Zhang et al., 2025; Du et al., 2025) under matched GPT-5 budgets. LEGO is the only system that succeeds on all 5 workloads with consistent speedups; among other agents, only CudaForge achieves a real speedup, and only on ResNet18 (1.75$\times$ on RTX 6000 Ada, 1.42$\times$ on H100). We further compare against TileLang (Wang et al., 2026; Cheng et al., 2025) (comparable performance across all five workloads; Table 5), and evaluate LEGO-optimized Qwen3-8B within HuggingFace Transformers (**15–21% lower time-to-first-token (TTFT)** than vLLM; Appendix F); using LEGO to optimize the attention operators that vLLM relies on reaches near-parity with FA2/FlashInfer at batch sizes $\geq 8$ (Appendix G).

### 4.4. Ablation Study: Dissecting the AND–OR Search

To validate the effectiveness of our design, we conduct a systematic ablation study on NVIDIA H100. We report *relative speedup over the best compiler baseline*, computed as the ratio of each configuration's speedup (vs. Eager) to TorchInductor's speedup (vs. Eager) under `max-autotune` mode. Values $> 1.0\times$ indicate improvement beyond the strongest compiler.

**Configurations.** We compare four variants that progressively remove key design choices:

- **Full LEGO**: The complete framework with unlimited recursive decomposition depth and speculative fusion.

- **Top-Down Only**: Disables bottom-up mutation. The system performs recursive decomposition but never attempts cross-boundary fusion.

- **Depth-1**: Restricts decomposition to a single level, where only top-level `nn.Module` children are optimized independently without further recursion.

- **Direct Codegen**: Generates a monolithic kernel for the entire target region without any decomposition.

**Analysis.** *(1) Fusion is essential for maximizing performance gains.* Removing fusion (Top-Down Only) substantially reduces speedup across all workloads. The degradation is most striking on mHC-MoE (11.29$\times$ $\to$ 1.20$\times$) and SD3-MMDiT (2.86$\times$ $\to$ 0.85$\times$), where the optimal execution plan requires fusing operators across module boundaries. Without bottom-up mutation, the system generates correct but fragmented kernels that miss cross-boundary data locality opportunities.

*Table 2.* Ablation study on NVIDIA H100. Speedups are relative to TorchInductor (`max-autotune`); values > 1.0× indicate improvement. × : codegen failed on most regions.

| Model | Full | Top-Down | Depth-1 | Direct |
|---|---|---|---|---|
| Mamba2-1.3B | 1.30× | 0.79× | × | × |
| SD3-MMDiT | 2.86× | 0.85× | × | × |
| mHC-MoE | 11.29× | 1.20× | 0.97× | 0.77× |
| ResNet18 | 1.02× | 1.01× | 0.93× | 0.94× |
| Qwen3-8B | 1.46× | 0.93× | × | × |

*(2) Deep recursion enables optimization of complex architectures.* Restricting to single-level decomposition (Depth-1) fails on Mamba2-1.3B, SD3-MMDiT, and Qwen3-8B. Without fine-grained recursive decomposition, the LLM cannot reliably generate correct kernels for complex intermediate modules. On simpler architectures like ResNet18, shallow decomposition remains viable but offers no improvement over the compiler.

*(3) Direct generation fails on non-trivial architectures.* Direct Codegen fails on three of five workloads and achieves suboptimal performance on the remainder. This validates that end-to-end optimization exceeds current LLM reasoning capacity without structural decomposition.

*(4) Components yield complementary benefits.* Deep recursion enables reliable code generation for complex graphs, while fusion recovers performance by exploiting cross-boundary optimization opportunities. The full framework's speedup is unattainable by either component alone.

### 4.5. Exploration Efficiency Analysis

To demonstrate the efficiency of the LEGO framework, we compare the exploration cost of our FusionTree search against a standard iterative LLM-based baseline. We report two key metrics: Exploration Speedup (reduction in wall-clock time) and Token Reduction Factor (reduction in total tokens consumed).

**Experimental Setup.** We define the baseline as a monolithic iterative LLM kernel generator, which attempts to generate kernels for the flattened graph directly using a `pass@10` sampling strategy combined with a `feedback@10` refinement loop. This baseline represents an unguided approach where the LLM is aggressively prompted to fix errors on the global context without structural decomposition. In contrast, LEGO employs a *Single-Shot FusionTree Search* ('pass@1') as described in Algorithm 1. Both methods continue exploration with internal errors until maximum feedback-based refinement is reached.

**Analysis.** As shown in Table 3, LEGO consistently outperforms the baseline. The gains come from three properties of

*Table 3.* **LEGO's Exploration Efficiency Comparison.** Speedup and Token Reduction (↑ is better) are calculated relative to the Monolithic Baseline (`pass@10` × `feedback@10`).

| Model | Speedup↑ | Token Reduction↑ |
|---|---|---|
| SD3-MMDiT | **1.94×** | **1.96×** |
| Mamba2-1.3B | **2.14×** | **7.02×** |
| mHC-MoE | **2.33×** | **4.93×** |
| Qwen3-8B | **2.47×** | **5.19×** |
| ResNet18 | **2.01×** | **2.48×** |

*Figure 4.* Final AND-OR tree status of Manifold-Constrained Hyper-Connections (mHC).

the FusionTree:

1. **Context Isolation:** Sub-agents operate on smaller subgraphs, producing correct code with fewer tokens. On Mamba, this yields $7.02×$ token reduction.
2. **Critical-Path Parallelism:** Sibling nodes are explored simultaneously; exploration time is bounded by tree depth (max 4), yielding $\sim 2×$ speedup.
3. **Bi-Directional Synergy:** Top-Down decomposition ensures validity on manageable subgraphs; Bottom-Up mutation recovers complex fusion patterns from verified components—combining reliability with high-performance locality.

### 4.6. Case Study: mHC — Zero-Day Optimization for Novel Operators

Manifold-Constrained Hyper-Connections (mHC) (Xie et al., 2025) employs $n$-stream residual expansion with doubly stochastic constraints enforced via iterative Sinkhorn

projection. This complexity makes it difficult for rigid compiler heuristics to identify optimal fusion boundaries.

As detailed in section 3 (and visualized in Figure 3), LEGO successfully decomposes mHC into verifiable sub-stages. Building on this decomposition, Figure 4 visualizes the final state of the AND-OR FusionTree. LEGO orchestrates a multi-agent search that explores `codegen` (leaf synthesis), `split` (structural decomposition), and `fuse` (speculative mutation) transitions. Each OR node retains the strategy with the minimal verified latency, which propagates bottom-up to determine the global plan. The mHC workload demonstrates LEGO's "Zero-Day" optimization capability—achieving **9.26×** speedup (H100) over eager execution without requiring any vendor-provided kernels.

## 5. Discussion

**Full-model recomposition.** LEGO measures end-to-end latency on the fully assembled model rather than summing per-sub-region speedups, avoiding systematic overestimation from cross-module synchronization and memory-layout overheads.

**Zero-day optimization.** LEGO synthesizes kernels from source code rather than selecting from a pre-built library, enabling acceleration of architectures for which no vendor kernel exists. The mHC-MoE workload (subsection 4.6) demonstrates this on the recently proposed mHC architecture (Xie et al., 2025).

**Complementarity with serving systems.** LEGO operates at the kernel-generation layer while serving engines such as vLLM (Kwon et al., 2023) optimize system-level concerns (KV-cache management, request scheduling, batching). The two are orthogonal and compose without conflict (Appendix F, Appendix G).

## 6. Conclusion

In this work, we presented LEGO, a hierarchical framework that resolves the fundamental tension between *global performance* and *exploration efficiency* in LLM-driven tensor optimization. By reformulating kernel generation as a **parallel multi-agent search** over a recursive **AND-OR FusionTree**, LEGO successfully synergizes the correctness of top-down structural decomposition with the high-performance locality of bottom-up speculative fusion. To the best of our knowledge, LEGO is the first framework to enable **scalable, efficient and performant** end-to-end GPU kernel generation, bridging the gap between local code synthesis and global model optimization.

## Acknowledgements

This work was supported in part by the National Key Research and Development Program of China (No. 2024YFE0204100), the National Natural Science Foundation of China (62302479, 62232015, 62090024), the China Postdoctoral Science Foundation (2023M733566), and the Australian Research Council (ARC) Grant (DP250104934). This work was also sponsored by CAAI-MindSpore Open Fund, developed on OpenI Community.

## Impact Statement

This paper presents work whose goal is to advance the field of machine learning. There are many potential societal consequences of our work, none of which we feel must be specifically highlighted here.

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

## A. Limitations

**Dependency on Frontier Model Capabilities.**  While LEGO reduces the search space complexity, the `Split` operation itself remains a reasoning-intensive task. Accurately parsing source code hierarchies and identifying optimal dependency cuts incurs non-trivial token consumption and latency. Consequently, the framework's current efficiency relies heavily on the capabilities of frontier Large Language Models (e.g., GPT-5). We observe that weaker models often struggle to maintain global structural coherence during decomposition, leading to sub-optimal cuts that significantly degrade the overall exploration efficiency of LEGO. To quantify this dependency, we compare GPT-5 against DeepSeek-V3 on two workloads (Table 4). GPT-5 produces substantial speedups on both, while DeepSeek-V3 fails on the complex mHC-MoE and achieves only marginal improvement on ResNet18. This reflects a capability trend rather than a fixed limitation—the framework improves with stronger backbone models.

*Table 4.* **Backbone model sensitivity** (H100, speedup over Eager).

| Backbone | mHC-MoE | ResNet18 |
|---|---|---|
| GPT-5 | 9.26× | 3.96× |
| DeepSeek-V3 | × | 1.12× |

**Future directions.**  Future work will explore heterogeneous model orchestration, dynamically routing complex structural reasoning to strong models while delegating local kernel synthesis to more lightweight, cost-effective models to optimize the total computational budget. We see several clear extensions suggested by our implementation: (i) a stronger `fuse` that performs multi-stage co-optimization (beyond recomposition), (ii) a stricter `patch_model` protocol to standardize safe module replacement and parameter transfer, and (iii) extending the runtime to additional accelerators by swapping backend clients while preserving the same AND–OR search interface.

## B. Details of Speculative Fusion Strategies

In this section, we provide the algorithmic details for the speculative fusion strategies introduced in Section 3.3. These strategies are executed in a bottom-up manner after the structural decomposition (Split) phase has yielded a set of valid child nodes.

The core intuition is to view the set of solved child sub-problems as a **context window** (visualized as dashed lines in our figures). The agent uses this context to propose a new, optimized AND-node that potentially "shorts" the graph, replacing a chain of separate kernels with a more efficient fused implementation.

### B.1. Aggressive Full-Fusion

The Full-Fusion strategy (Alg. 2) attempts to collapse the entire decomposed subgraph back into a single monolithic kernel. This essentially tries to convert a *Structural AND-node* (composed of multiple kernel calls) into a single *Terminal AND-node*.

---

**Algorithm 2** Speculative Full-Fusion

---

1: **Input:** Solved Child OR-Nodes $\{v_1, v_2, \ldots, v_k\}$
2: **Output:** Candidate AND-Node $u_{full}$ (or null if failure)
3: *// 1. Context Collection (The Dashed Line)*
4: *// Flatten the subgraph to retrieve all atomic operators*
5: $OpGraph \leftarrow$ Flatten($\bigcup_{i=1}^{k} v_i$.BestPlan)
6: *// 2. Monolithic Compilation*
7: *// Attempt to generate a single kernel for the entire graph*
8: $SourceCode \leftarrow$ LLM_GenKernel($OpGraph$)
9: $u_{full} \leftarrow$ Compile($SourceCode$)
10: *// 3. Validation*
11: **if** $u_{full}$ compiles successfully **and** Cost($u_{full}$) $< \sum$ Cost($v_i$) **then**
12:     **return** $u_{full}$ *// Success: Global locality optimization*
13: **else**
14:     **return null** *// Failure: Resource exhaustion or performance regression*
15: **end if**

---

## B.2. Selective Partial-Fusion

The Partial-Fusion strategy (Alg. 3) is more sophisticated. It acknowledges that full fusion is not always optimal. Instead, it creates a **Hybrid AND-node** that mixes a newly generated fused kernel with pointers to existing, already-solved sibling nodes.

Crucially, this algorithm requires **Structural Reuse**: solid edges are created pointing to the existing OR-nodes (e.g., $v_{keep}$) to avoid re-compiling unchanged parts of the graph.

---

**Algorithm 3** Speculative Partial-Fusion

---

1: **Input:** Solved Child OR-Nodes $\{v_1, \ldots, v_k\}$, Dependency Graph $G$
2: **Output:** Candidate AND-Node $u_{part}$
3: *// 1. Semantic Partitioning (LLM Decision)*
4: *// Prompt LLM to identify the "Diamond" or "Hotspot" sub-region*
5: $S_{fuse}, S_{keep} \leftarrow$ LLM_SelectPartition($G, \{v_1, \ldots, v_k\}$)
6: *// e.g., Fuse {Pre, Res}, Keep {F, Post}*
7: *// 2. Fusion Synthesis (The Dashed Line)*
8: *// Generate kernel only for the selected subset*
9: $OpGraph_{sub} \leftarrow$ Flatten($S_{fuse}$)
10: $Kernel_{new} \leftarrow$ Compile(LLM_GenKernel($OpGraph_{sub}$))
11: *// 3. Hybrid Node Construction (Structural Reuse)*
12: $u_{part} \leftarrow$ NewANDNode()
13: *// Edge 1: Point to the new fused kernel*
14: $u_{part}$.children.add($Kernel_{new}$)
15: *// Edge 2..N: Reuse existing solved OR-nodes (Solid Lines)*
16: **for all** $v \in S_{keep}$ **do**
17:     $u_{part}$.children.add($v$) {Reusing compiled result $O(1)$}
18: **end for**
19: *// 4. Cost Aggregation*
20: Cost($u_{part}$) $\leftarrow$ Cost($Kernel_{new}$) $+ \sum_{v \in S_{keep}}$ Cost($v$)
21: **return** $u_{part}$

---

# C. Implementation Primitives

In this section, we formally define the algorithmic primitives used in the LEGO framework.

### C.1. The Interference-Free Profiling System

The `Measure` primitive serves as the ground-truth verifier for our Multi-Agent exploration. Since the AO-graph exploration spawns purely asynchronous agents that concurrently generate and compile kernels, naive profiling would lead to severe **resource interference** (e.g., L2 cache contention, SM occupancy fluctuations), resulting in noisy and unreliable reward signals.

To mitigate this, we architect a centralized **GPU Profiling Service**. The system operates on a Producer-Consumer model:

- **Producers (Agents):** Upon successful compilation, agents encapsulate the binary into a profiling task and push it into a global **FIFO Job Queue**.

- **Consumer (GPU Pool):** A dedicated worker manages the GPU resource pool. We enforce strict mutual exclusion using a **System Semaphore** (or Global Lock). Only one kernel is allowed to occupy the device at any given timestamp.

Algorithm 4 details this serialized measurement pipeline, ensuring that every latency number ($t$) used in the Bellman updates represents the deterministic, interference-free performance of the kernel.

---

**Algorithm 4** Primitive: Measure with Global Serialization

---

1: **Input:** Candidate Implementation $\pi$ (Source Code)
2: **Output:** Latency $t \in \mathbb{R}^+ \cup \{\infty\}$
3: *// 1. Compilation & Functional Verification*
4: $executable \leftarrow \text{Compile}(\pi)$
5: **if** compilation fails **or** runtime error **then**
6:     **return** $\infty$
7: **end if**
8: *// 2. Numerical Precision Check*
9: $y_{pred} \leftarrow \text{Execute}(executable, \text{RandomInputs})$
10: $y_{ref} \leftarrow \text{Execute}(\text{PytorchEager}, \text{RandomInputs})$
11: **if** $\text{MaxDiff}(y_{pred}, y_{ref}) > \epsilon_{tol}$ **then**
12:     **return** $\infty$           *// Fails accuracy requirement*
13: **end if**
14: *// 3. Serialized Profiling (Interference-Free)*
15: *// Submit task to the Global FIFO Queue to compete for GPU resources*
16: $task \leftarrow \text{NewTask}(executable, \text{warmup} = 10, \text{repeat} = 100)$
17: GlobalQueue.push($task$)
18: *// Block agent until the GPU semaphore is acquired*
19: **wait** GlobalGPUSemaphore
20: *// Critical Section: Exclusive Hardware Access*
21: $t \leftarrow \text{ProfileKernel}(task)$
22: **signal** GlobalGPUSemaphore
23: **return** $t$

---

## D. Compiler Baseline: TileLang and Optimization Cost

We compare against TileLang (Wang et al., 2026; Cheng et al., 2025), a state-of-the-art tile-level DSL compiler. We manually implement and tune TileLang kernels for each model and integrate with `torch.compile(max-autotune)`.

*Table 5.* **TileLang vs. LEGO** (RTX 6000 Ada, speedup over Eager, both with `torch.compile(max-autotune)`). **Bold**: best per row.

| Model | TileLang | LEGO |
|---|---|---|
| ResNet18 | 5.87× | **6.00×** |
| Qwen3-8B | **2.25×** | 2.21× |
| SD3-MMDiT | 3.94× | **4.09×** |
| Mamba2-1.3B | 2.12× | **2.18×** |
| mHC-MoE | 13.01× | **13.48×** |

LEGO's automatically synthesized kernels achieve performance comparable to manually engineered TileLang kernels across all workloads. TileLang leads on Qwen3-8B due to hand-tuned GQA attention, but requires manual per-region kernel engineering. LEGO's Triton kernels are *compiler-transparent*, composing with TorchInductor's cross-operator optimizations, whereas TileLang kernels (TVM FFI) are opaque to the compiler.

## E. Source of Speedup Analysis

To quantify the contribution of LLM-synthesized kernels vs. vendor library calls, we profile GPU-time share under `LEGO + torch.compile(max-autotune)` on RTX 6000 Ada (Table 6).

*Table 6.* **GPU-time share**: LEGO-synthesized vs. vendor kernels.

| Model | LEGO | Vendor | Key synthesized kernels |
|---|---|---|---|
| Qwen3-8B | 98% | 2% | Fused attention (norm+RoPE+online-softmax), RMSNorm+projection |
| SD3-MMDiT | 68% | 32% | Fused joint-attention kernel |
| Mamba2-1.3B | 55% | 45% | Fused SSD scan (entire SSM recurrence in 1 kernel) |
| mHC-MoE | 49% | 51% | Fused grouped-GEMM (routing+dispatch+expert SwiGLU) |
| ResNet18 | 53% | 47% | Fused norm+activation, residual+pool |

LEGO synthesizes kernels where cross-boundary fusion and algorithmic transformation yield the largest gains: $O(S^2)$-materialization attention in Qwen3/SD3, the 64-expert serial dispatch in mHC-MoE, and the multi-step SSD recurrence in Mamba. Vendor libraries (cuBLAS, cuDNN) are deliberately retained for single-operator peak throughput.

## F. Integration with HuggingFace Transformers

We evaluate LEGO-optimized Qwen3-8B inference within the HuggingFace Transformers framework (BF16, batch size 1, RTX 6000 Ada, real Qwen3-8B weights) and compare prefill latency against three baselines: plain Eager execution, PyTorch SDPA, and vLLM (Table 7).

*Table 7.* **HuggingFace Transformers prefill latency** (ms, Qwen3-8B, $B$=1, BF16, RTX 6000 Ada). [†]OOM at this sequence length.

| Prompt | Eager | SDPA | LEGO | vLLM (ref) |
|---|---|---|---|---|
| 512 | 103 | 86 | **67** | 83 |
| 1024 | 220 | 172 | **130** | 165 |
| 2048 | 501 | 343 | **265** | 319 |
| 4096 | 1264 | 750 | **524** | 618 |
| 8192 | —[†] | 1660 | **1239** | 1481 |

LEGO achieves **15–21% lower TTFT than vLLM** across all prompt lengths, and **1.5–2.4× lower latency than Eager** where Eager is feasible. SDPA provides a kernel-level attention speedup but leaves surrounding operators unoptimized; at $P$=4096 SDPA (750 ms) is still 43% slower than LEGO (524 ms). At $P$=8192, naive Eager execution is infeasible (OOM), SDPA regresses to 1660 ms, while LEGO sustains 1239 ms—16% faster than vLLM (1481 ms).

# G. Comparison with LLM Serving Engines

We use LEGO to synthesize the attention operators that vLLM (Kwon et al., 2023) relies on (v0.15, RTX 6000 Ada, FP16, real Qwen3-8B weights) and compare them against vLLM's default FlashAttention-2 (FA2) and FlashInfer kernels. We measure prefill latency (time-to-first-token, TTFT; Table 8) and decode throughput (time-per-output-token, TPOT; Table 9) across batch sizes $B \in \{1, 4, 8, 16\}$ and prompt lengths $P \in \{512, 1024, 2048, 4096\}$.

*Table 8.* **vLLM prefill latency** (ms, Qwen3-8B, output_len=1). Three attention-kernel implementations within the same vLLM v0.15 engine.

| B | Prompt | +FA2 | +LEGO | +FlashInfer | Best |
|---|--------|------|-------|-------------|------|
| 1 | 512 | **82** | 83 | 84 | FA2 |
| 1 | 1024 | **161** | 165 | **161** | FA2 |
| 1 | 2048 | **312** | 327 | **312** | FA2 |
| 1 | 4096 | **608** | 673 | 609 | FA2 |
| 4 | 512 | 326 | 328 | **324** | FI |
| 4 | 1024 | 638 | 653 | **637** | FI |
| 4 | 2048 | **1327** | 1398 | 1334 | FA2 |
| 4 | 4096 | 2763 | 3048 | **2749** | FI |
| 8 | 512 | 617 | 616 | **608** | FI |
| 8 | 1024 | **1252** | 1306 | 1267 | FA2 |
| 8 | 2048 | **2770** | 2909 | **2770** | FA2 |
| 8 | 4096 | 5682 | 6276 | **5627** | FI |
| 16 | 512 | 1253 | 1263 | **1242** | FI |
| 16 | 1024 | 2682 | 2762 | **2676** | FI |
| 16 | 2048 | 5609 | 5859 | **5574** | FI |
| 16 | 4096 | 11416 | 12584 | **11328** | FI |

**Prefill latency (TTFT).** At $B{=}1$, FA2's hand-written C++ kernel leads (1–10% faster than LEGO); FlashInfer ties FA2 at $P{\leq}2048$. At $B{\geq}4$, FlashInfer's paged-KV implementation dominates prefill. LEGO's Triton backend stays within 1–10% of the best backend across all 16 configurations—competitive given that it is entirely LLM-synthesized.

*Table 9.* **vLLM E2E TPOT** (ms/token, Qwen3-8B, gen=128). Same three attention-kernel implementations.

| B | Prompt | +FA2 | +LEGO | +FlashInfer | Best |
|---|--------|------|-------|-------------|------|
| 1 | 512 | **18.5** | 20.5 | 19.1 | FA2 |
| 1 | 1024 | **18.7** | 20.6 | 19.4 | FA2 |
| 1 | 2048 | **19.1** | 20.7 | 19.6 | FA2 |
| 1 | 4096 | **19.8** | 21.1 | 20.2 | FA2 |
| 4 | 512 | **19.5** | 21.1 | 20.4 | FA2 |
| 4 | 1024 | **20.3** | 21.4 | 20.9 | FA2 |
| 4 | 2048 | **21.2** | 21.3 | 21.5 | FA2 |
| 4 | 4096 | 22.7 | **22.5** | 22.9 | LEGO |
| 8 | 512 | **20.6** | 21.5 | 21.2 | FA2 |
| 8 | 1024 | **21.3** | 21.6 | 21.8 | FA2 |
| 8 | 2048 | 22.3 | **22.1** | 22.8 | LEGO |
| 8 | 4096 | **25.3** | 25.3 | 25.7 | FA2 |
| 16 | 512 | 21.9 | **21.9** | 22.4 | LEGO |
| 16 | 1024 | 22.9 | **22.8** | 23.5 | LEGO |
| 16 | 2048 | 25.3 | **25.2** | 25.7 | LEGO |
| 16 | 4096 | 31.7 | 31.4 | **31.3** | FI |

**Decode throughput (TPOT).** At batch size 1, decode is dominated by weight-loading GEMMs (∼84% of TPOT is cuBLAS), leaving the attention kernel with only 4–16% of wall time; FA2's C++ implementation retains a ∼1–2 ms edge. At batch size 16, LEGO matches or slightly outperforms FA2 across all prompt lengths, and at batch size 8 it reaches parity for longer prompts (≥2048); larger batches make the attention kernel a bigger fraction of TPOT and amortize Triton's launch overhead. FlashInfer is strongest at batch size 16, prompt length 4096.

**Complementarity.**   LEGO and vLLM address orthogonal optimization layers. The HF+LEGO+compile integration (Appendix F) maximizes prefill throughput via full-graph kernel fusion. Pairing LEGO's optimized attention operators with vLLM retains its PagedAttention and CUDAGraph-based batching for decode throughput. Both paths are viable simultaneously: LEGO's optimized operators can be integrated into any framework exposing a modular attention interface.

## H. Correctness Validation Pipeline

LEGO enforces a three-stage correctness pipeline before any candidate is eligible for profiling:

1. **Code well-formedness**: Pyright/LSP-based static analysis verifies type correctness and API usage before runtime evaluation.

2. **Submodule validation**: Each recursively decomposed submodule is numerically checked against the reference (`atol=1e-2, rtol=1e-2`), aligned with prior work (Ouyang et al., 2025; Dong et al., 2025).

3. **Model-level validation**: For deployment-critical workloads (Qwen3-8B), we verify end-to-end behavioral correctness under real pretrained weights—the first 200 decode tokens must match the reference model exactly.

This hierarchical validation prevents cascading numerical errors and ensures that candidates are not only locally correct but globally coherent. Recent work (Lange et al., 2025) demonstrates that single-configuration verification can permit artificial speedups of up to $120\times$, reinforcing the importance of multi-level validation as implemented in LEGO.

## I. Prompts and Optimization Traces

The three core actions in the AND-OR FusionTree—SPLIT, GENERATE, and FUSE—each correspond to a separate prompt. The versions shown below have been condensed for presentation; the full prompts contain more detailed descriptions, environment setup, error-handling instructions, and interface contracts.

### I.1. Split Prompt

```
You are a performance analysis agent. Your task is to decompose the
following PyTorch model into independently-optimizable sub-regions by
aligning with its natural source-code module boundaries.

[MODEL CODE]

=== Decomposition principle ===
nn.Module boundaries are natural split points because each module
encapsulates a self-contained computation with a well-defined interface.
A valid split requires: (1) no shared mutable state between sub-regions,
(2) the sub-region's forward signature fully specifies its data contract.

=== Protocol ===

Step 1 | Profile.
  model = Model(*get_init_inputs()).eval().cuda()
  inputs = [x.cuda() if isinstance(x, torch.Tensor) else x
            for x in get_inputs()]
  with torch.profiler.profile(
      activities=[ProfilerActivity.CUDA]) as p:
    with torch.inference_mode(): model(*inputs)
  print(p.key_averages().table(sort_by="cuda_time_total", row_limit=20))

... (warm-up protocol, multi-run statistics, stack-trace attribution)

Step 2 | Enumerate the nn.Module tree and map profiler events to modules.
  for name, mod in model.named_modules():
      if not list(mod.children()):
          print(name, type(mod).__name__)
  Also inspect forward() for non-module computations (loops, inline
  math) that may not appear as named sub-modules but still consume
  significant CUDA time.

Step 3 | For each sub-region (module or identified code block), record:
  - module path, bottleneck category (compute-bound? memory-bound?
    launch-overhead-bound?), exact interface (input/output names,
    shapes, dtypes), and whether isolated optimization is safe.
```

```
Step 4 | Decide action for each sub-region:
  Codegen        : the sub-region maps to a single kernel-sized unit
                   that can be directly optimized as one piece.
  recurse-Split  : the sub-region contains heterogeneous stages with
                   independent data-flow patterns | decompose further.

Report a decomposition plan in this format:
  SPLIT PLAN
    Sub-regions (prioritized by CUDA time share):
      [module.path]:
        bottleneck: <why it is slow>
        interface:  <input shapes/dtypes → output shapes/dtypes>
        action:     <Codegen | recurse-Split>

... (environment setup, sandbox constraints, output formatting rules)
```

## I.2. Generate Prompt

```
You are working in an isolated sandbox. Do NOT read any file outside
this directory. Your task: optimize one isolated sub-region.

[ISOLATED SUB-REGION CODE + INTERFACE SPECIFICATION]

... (sandbox setup, file access restrictions, dependency versions)

=== Hard interface contract ===
The verification harness does STRICT matching:
  1. Class name, __init__ signature (all parameters + defaults), and
     forward signature must be IDENTICAL to the reference.
  2. All state_dict keys must match: same attribute names, same shapes.
     Do NOT rename self.x → self._x or add new nn.Parameter/register_buffer
     entries. If you stack or cache weights for kernel efficiency, store
     them as plain Python attributes (not Parameter, not persistent buffer)
     so they are invisible to state_dict.
  3. Numerical tolerance: atol=1e-2, rtol=1e-2 against the fp16 reference.

=== Optimization protocol ===
Write model_optimized.py. General approach:
  - For compute-intensive bottlenecks with non-standard access patterns
    (irregular reductions, iterative algorithms, tiled attention): write
    a @triton.jit kernel.
  - For large, regular GEMMs and convolutions: let torch.compile handle
    them; wrap only the surrounding control flow.
  - Prefer in-place operations to avoid extra tensor allocations.

Required imports at the top of model_optimized.py:
  import torch, torch.nn as nn, torch.nn.functional as F
  import triton
  import triton.language as tl

Triton kernel writing rules:
  - grid: match the natural parallelism (one program per output row, per
    batch element, or a fixed NUM_SM-wide persistent grid for irregular work)
  - Declare block sizes as tl.constexpr; pass them at launch time
  - Boundary-safe loads: tl.load(ptr + offs, mask=offs < N, other=0.)
  - Accumulate reductions in fp32; store results in fp16/bf16
  - Use tl.dot(a, b) for in-tile matmul; tl.math.exp2(x*1.44269504) for exp
  - Do not use torch._grouped_mm or other prebuilt grouped matmul APIs

... (memory coalescing rules, shared memory usage, warp-level primitives)

Common Triton mistakes to avoid:
  - tl.tanh does NOT exist → use tl.math.tanh(x) instead
  - tl.sigmoid does NOT exist → compute 1.0/(1.0+tl.math.exp(-x*1.44269504))
  - tl.exp(x) is slower than tl.math.exp2(x * 1.44269504) on modern GPUs
  - Kernel launch uses square brackets: kernel[(grid,)](args, BLOCK=128)
    NOT kernel(grid, args); the grid must be a tuple inside brackets
  - Block sizes used as tl.constexpr must be compile-time constants
    (powers of 2 work best); do not compute them from tensor shapes at runtime
  - tl.load / tl.store require contiguous pointer arithmetic; if the input
    tensor is non-contiguous, call .contiguous() before passing its data_ptr()

Iterate in this fixed order:
  write model_optimized.py
  → python run_verify.py model_optimized.py   # fix until PASS
  → python run_bench.py  model_optimized.py   # record speedup
  → diagnose bottleneck, improve, repeat
Stop when marginal gain < 1% across two consecutive iterations or no
further optimization idea remains.
```

```
Report: final speedup, correctness (PASS/FAIL + max_diff), and a brief
description of each Triton kernel written (grid, what it replaces).

...  (additional Triton idioms, debugging heuristics, autotuning guidance)
```

## I.3. Fuse Prompt

```
The following sub-regions have each been independently optimized and
verified. Now look for cross-boundary optimization opportunities.

  Sub-region A: [name, interface, latency after Codegen]
  Sub-region B: [name, interface, latency after Codegen]

...  (interface contract inherited from Generate, sandbox constraints)

Step 1 | Profile the combined implementation to find remaining overhead:
  python run_bench.py model_combined.py --profile
  Identify elementwise or normalization ops that account for > 5% of
  remaining runtime.

Step 2 | Check each cross-boundary fusion candidate:

  Candidate 1 | residual-add + normalization at a module boundary:
    Pattern: out_A = sub_region_A(x); x_new = residual + out_A; y = norm(x_new)
    Cost: x_new is written to HBM and immediately reloaded for norm.
    Fusion: one @triton.jit kernel reads residual + out_A, writes x_new
    in-place, and returns norm(x_new) in the same pass.

  Candidate 2 | batching independent linear projections:
    Pattern: y1 = proj_1(x); y2 = proj_2(x); ...  (same input tensor)
    Fusion: stack weights → one GEMM; use strided views to split output.

  Candidate 3 | elementwise epilogue chain of an existing kernel:
    If one or more elementwise/normalization ops immediately follow a
    kernel that still has the output in L2, fold them into that
    kernel's output write loop.

...  (additional fusion patterns: broadcast elimination, layout transposition)

Step 3 | For each candidate pursued:
  (a) Estimate savings: bytes_saved / device_bandwidth > target_threshold
  (b) Implement the fused version; verify: python run_verify.py model_fused.py
  (c) Benchmark: python run_bench.py model_fused.py
  (d) Accept if latency(fused) < latency(A) + latency(B); else revert.

Report: each fusion attempted, whether it was accepted or reverted,
and the latency delta.

...  (HBM bandwidth estimation templates, layout-mismatch detection rules)
```

## I.4. Hierarchical Decomposition Trace: mHC-MoE (SPLIT → SPLIT → GENERATE)

This case study walks through two successive SPLIT actions followed by a GENERATE action, illustrating how LEGO's hierarchical decomposition exposes deeply nested optimization opportunities.

**Level-1 Split: whole model → three sub-regions.** The mHC-MoE model (Xie et al., 2025) combines a Manifold-Constrained Hyper-Connection layer with a Mixture-of-Experts block. LEGO applies the Split prompt to the full model code and proposes decomposing it into three independently optimizable sub-regions based on data-flow boundaries:

```python
from pre_res.model import Model as PreResModel
from moe.model   import Model as MoEModel
from post.model  import Model as PostModel

class Model(nn.Module):
    def __init__(self, hidden_dim, expansion_rate=4,
                 sinkhorn_iters=20, sinkhorn_eps=1e-8, ...):
        self.pre_res = PreResModel(hidden_dim, ...)   # coefficient gen + pre-mapping
        self.moe     = MoEModel(hidden_dim, ...)      # MoE layer function F
        self.post    = PostModel(hidden_dim, ...)     # residual + post-mapping

    def forward(self, u):
```

```
        h_in_norm, H_post, H_res = self.pre_res(u)    # produces routing matrices
        h_out = self.moe(h_in_norm)
        return self.post(u, H_post, H_res, h_out)
```

Each sub-region (`pre_res`, `moe`, `post`) is a valid `nn.Module` with a well-defined interface, enabling parallel optimization by independent agents.

**Level-2 Split: `pre_res` → identifies `sinkhorn` bottleneck.**    The Split prompt is applied recursively to `pre_res`. The agent identifies `pre_res.sinkhorn()` as the optimization target due to its iterative loop of sequential kernel launches. The agent reports:

```
Bottleneck: Model.sinkhorn(log_M) -> M
- Iterative normalization loop with many sequential kernel launches
- Safe to optimize in isolation: no external state mutation
- Proposed action: Generate a fused Triton kernel
```

The identified sub-region is then passed to a GENERATE agent. The following specification fills the [ISOLATED SUB-REGION CODE + INTERFACE SPECIFICATION] placeholder in the Generate prompt (subsection I.2):

```
Sub-region: Model.sinkhorn(log_M) -> M
Input:  log_M  [B, N, N]  fp32
Output: M      [B, N, N]  fp32

Source code:
    def sinkhorn(self, log_M):
        M = torch.exp(log_M - log_M.amax(dim=(-2,-1), keepdim=True))
        for _ in range(self.sinkhorn_iters):
            M = M / (M.sum(dim=-2, keepdim=True) + self.sinkhorn_eps)
            M = M / (M.sum(dim=-1, keepdim=True) + self.sinkhorn_eps)
        return M

Goal: reduce its execution time while preserving numerical output.
```

**GENERATE: Triton kernel result.**

```python
# LEGO output: all Sinkhorn iterations fused into one kernel, M stays in registers
@triton.jit
def _sinkhorn_kernel(inp_ptr, out_ptr, stride_b, stride_i, stride_j,
                     B, N: tl.constexpr, n_iters: tl.constexpr,
                     eps: tl.constexpr):
    b = tl.program_id(0)                  # one program per batch element
    i = tl.arange(0, N); j = tl.arange(0, N)
    base = inp_ptr + b * stride_b
    log_M = tl.load(base + i[:,None]*stride_i + j[None,:]*stride_j)
    max_val = tl.max(tl.max(log_M, axis=1), axis=0)
    M = tl.math.exp2((log_M - max_val) * 1.4426950408889634)  # fp32, in regs

    for _ in range(n_iters):              # all iterations inside the kernel
        col_sum = tl.sum(M, axis=0)
        M = M / (col_sum[None,:] + eps)
        row_sum = tl.sum(M, axis=1)
        M = M / (row_sum[:,None] + eps)

    out_base = out_ptr + b * stride_b  # store final result once
    tl.store(out_base + i[:,None]*stride_i + j[None,:]*stride_j, M)
```

Correctness: PASS (max_diff$<10^{-5}$, verified against the Python-loop reference at the same shapes and iteration count). The kernel fuses all normalization iterations into a single launch: the $[N, N]$ routing matrix lives in registers throughout and is written to HBM exactly once, eliminating all intermediate round-trips.

### I.5. Case Study: Qwen3-8B Cross-Layer Fusion (FUSE)

**Cross-boundary opportunity.** After GENERATE independently optimizes the attention and MLP sub-regions of a transformer decoder, the layer forward loop contains redundant HBM round-trips at each sub-region boundary:

```python
# Standard PyTorch: two separate kernel launches per layer boundary
for layer in self.layers:
    x = x + layer.self_attn(layer.input_layernorm(x))        # attn → residual add → HBM write
    x = x + layer.mlp(layer.post_attention_layernorm(x))     # MLP → residual add → HBM write
    # next layer: input_layernorm immediately re-reads x from HBM
```

The FUSE agent identifies that the residual add at the end of each sub-region and the normalization at the start of the next sub-region access the same tensor sequentially, with a wasteful HBM round-trip in between.

**FUSE-generated kernel.** The following kernel is produced by the FUSE agent for the Qwen3-8B optimization (Table 1):

```python
@triton.jit
def _add_rmsnorm_kernel(
    x_ptr, delta_ptr, out_ptr, weight_ptr, rows,
    D: tl.constexpr, eps: tl.constexpr, BLOCK_D: tl.constexpr,
):
    """One HBM pass: load x+delta, update x in-place, return rmsnorm(x+delta)."""
    row = tl.program_id(0)
    if row >= rows:
        return
    offs = tl.arange(0, BLOCK_D)
    xi = tl.load(x_ptr      + row * D + offs).to(tl.float32)
    di = tl.load(delta_ptr + row * D + offs).to(tl.float32)
    xd = xi + di                                  # residual add (fp32)
    var = tl.sum(xd * xd) / D
    inv_rms = 1.0 / tl.math.sqrt(var + eps)
    tl.store(x_ptr   + row * D + offs, xd.to(tl.float16))   # update residual in-place
    w = tl.load(weight_ptr + offs).to(tl.float32)
    tl.store(out_ptr + row * D + offs, (xd * inv_rms * w).to(tl.float16))
```

The fused forward loop threads this kernel across layer boundaries, eliminating one HBM round-trip per layer:

```python
# Fused forward: cross-layer add+norm in one pass per boundary
x_normed = triton_rmsnorm(x, self.layers[0].input_layernorm.weight, ...)
for i, layer in enumerate(self.layers):
    attn_out = layer.self_attn(x_normed, ...)
    # Fused: x += attn_out; return post_attention_layernorm(x)
    x_normed2 = triton_add_rmsnorm(x, attn_out,
                                   layer.post_attention_layernorm.weight, ...)
    mlp_out = layer.mlp(x_normed2)
    # Fused across layer boundary: x += mlp_out; return next_layer's input_layernorm(x)
    next_norm_weight = (self.layers[i+1].input_layernorm.weight
                        if i+1 < len(self.layers) else self.norm.weight)
    x_normed = triton_add_rmsnorm(x, mlp_out, next_norm_weight, ...)
```

This fusion is applied in the Qwen3-8B optimization. The full model achieves **2.21×** speedup over PyTorch eager (Table 1), with the FUSE-produced kernel contributing to the post-GENERATE residual non-GEMM overhead reduction.

