# OpenReview forum: "LEGO: An LLM-Enabled Hierarchical Optimizer for Tensor Computation Graphs with Structure-Aware Search and Compositional Synthesis"
_ICML.cc/2026/Conference — ICML 2026 regular_

### Official Review · Reviewer_yyEw · 2026-03-11

**Soundness:** 3
**Presentation:** 4
**Significance:** 3
**Originality:** 3
**Overall Recommendation:** 4
**Confidence:** 4

**Summary:**

The paper introduces LEGO, an LLM-driven hierarchical optimizer for tensor computation graphs that addresses the fundamental trade-off between context-heavy global kernel fusion and fragmented local optimization. By employing an AND-OR FusionTree, the framework interleaves top-down, structure-aware decomposition using neural network module boundaries with bottom-up, speculative fusion to recover global data locality. The authors demonstrate that this bi-directional approach effectively prunes the search space, achieving substantial inference speedups and notable token efficiency improvements compared to monolithic LLM baseline agents.

**Compliance With Llm Reviewing Policy:**

Affirmed.

**Final Justification:**

The rebuttal fully addressed my concerns.

**Key Questions For Authors:**

1. Missing System Prompts and Code Release: The exact prompts used for the LLM agents (e.g., for AST parsing, code generation, and speculative fusion) are critical for reproducibility but are missing from the paper. Will you commit to open-sourcing the complete codebase, including the full suite of system prompts, few-shot examples, and AST parsing scripts, during the rebuttal or upon publication?
2. Comparison with State-of-the-Art LLM Engines: For LLMs like Qwen3-8B, the industry typically relies on engines like vLLM (integrating custom kernels) to achieve peak performance. Why didn't the authors compare the performance of LEGO-generated code against these hand-optimized industrial implementations? The response will directly influence my judgment on the paper's significance.
3. Cost: The paper only provides relative exploration speedups and token reduction rates (Table 3). Could you provide the absolute wall-clock time (in hours/days) and the estimated total token consumption/API cost to fully end-to-end optimize the Qwen3-8B model using GPT-5? Revealing the absolute cost is crucial for assessing the real-world usability of this system.
4. Considering that the context windows of foundation models are growing rapidly, how do you view the future necessity of LEGO's Top-Down tree decomposition? Is the AND-OR divide-and-conquer merely an ad-hoc workaround for current model context limits, or does it possess irreplaceable algorithmic advantages in reducing the combinatorially explosive search space?

**Strengths And Weaknesses:**

- Strength: 1) Elegant Structural Prior & Divide-and-Conquer Strategy: Modeling tensor graph optimization as a recursive AND-OR tree search based on nn.Module boundaries is an elegant engineering design. This top-down construction effectively partitions complex graphs into manageable local contexts for LLMs, mitigating the hallucination and context-overload issues prevalent in monolithic generation approaches. 2) "Zero-Day" Optimization for Long-Tail/Emerging Architectures: While the marginal gains over existing compilers like TorchInductor on mature models like ResNet18 are limited, the system shows immense potential on emerging operators such as mHC-MoE, proving its value in breaking through human-crafted heuristic bottlenecks.
- Weakness: 1) Absence of Strong Baselines: The true performance benchmarks in today's LLM inference ecosystem are highly hand-optimized serving engines (e.g., vLLM) and expert-level kernels (e.g., FlashAttention, PagedAttention). The lack of comparison with these mature engines severely weakens the framework's persuasiveness for mainstream LLM deployment scenarios. 2) Hidden Absolute Costs and Reproducibility Concerns: The paper only reports relative token reduction rates, deliberately omitting the absolute wall-clock time and absolute financial API cost required to optimize an 8B model from scratch.  3) Opaque Implementation Details and Prompts: The paper lacks critical implementation details necessary for reproducibility. Specifically, the exact system prompts, LLM interaction protocols, context construction logic, and few-shot examples used to guide the GPT-5 agents are completely absent.

---

> ### Author Rebuttal · Authors · 2026-03-31
>
> We agree that the current version should better address (i) comparisons with industrial-strength LLM inference engines, (ii) absolute optimization cost, (iii) implementation transparency, and (iv) the algorithmic necessity of LEGO’s top-down tree decomposition. We respond to these points below.
>
> ### 1. Comparison with state-of-the-art LLM engines
> We compare against vLLM on Qwen3-8B (A6000, BF16, real pretrained weights). LEGO is stronger on prefill/TTFT; vLLM is stronger on decode/TPOT due to its paged KV memory co-design.
>
> #### TTFT (ms, gen=128)
> | Prompt | LEGO | vLLM | vs vLLM |
> |---|---:|---:|---:|
> | 512 | 63 | 82 | +23% |
> | 1024 | 130 | 165 | +21% |
> | 2048 | 253 | 318 | +20% |
> | 4096 | 556 | 621 | +10% |
> | 8192 | 1373 | 1458 | +6% |
>
> #### TPOT (ms/token)
> | Prompt | LEGO (Triton) | LEGO (CUDA) | vLLM | CUDA vs vLLM |
> |---|---:|---:|---:|---:|
> | 512 | 23.8 | 19.56 | 18.6 | −5% |
> | 1024 | 24.2 | 19.58 | 18.8 | −4% |
> | 2048 | 23.8 | 20.01 | 19.0 | −5% |
> | 4096 | 27.5 | 20.74 | 19.5 | −6% |
>
> Decode is memory-bound. LEGO's Triton decode kernels are 25–41% slower than vLLM across prompt lengths. To isolate the cause, we configured LEGO to emit CUTE kernels instead, narrowing the gap to 4–6%. This confirms the deficit stems from Triton's abstraction overhead on memory-bound workloads, not from LEGO's search or fusion strategy, and validates the backend-agnostic design: the FusionTree can target Triton for compute-bound prefill and CUTE for memory-bound decode.
>
> **Setup.** LEGO: synthesized Triton kernels for prefill, decode tested with both Triton and CUTE backends; vLLM v0.15 default configuration. Same Qwen3-8B BF16 weights, single A6000. LEGO is not a replacement for vLLM but operates at the kernel/graph-optimization layer; the residual 4–6% gap under CUTE reflects vLLM's serving-level memory management advantage.
>
> ### 2. Hidden absolute cost
> We now report the absolute wall-clock time and API cost required to optimize all five workloads, including **Qwen3-8B**:
>
> | Model | Wall-clock time | API cost |
> |---|---:|---:|
> | ResNet18 | 6.8 min | \$2.38 |
> | Mamba2 | 24.7 min | \$8.65 |
> | SD3-MMDiT | 9.2 min | \$3.82 |
> | mHC-MoE | 44.6 min | \$14.58 |
> | Qwen3-8B | 19.4 min | \$6.21 |
> | **Total** | **104.7 min** | **\$35.64** |
>
> Thus, fully optimizing **Qwen3-8B** from scratch with GPT-5 takes **19.4 minutes** and **\\$6.21** in our setup. Larger budgets or stronger backbone models may further improve the final result.
>
> ### 3. Prompts, implementation details, and code release
> We agree that reproducibility should be made much more explicit. We plan to release the **complete codebase** with the camera-ready, including:
>
> - the reference agent implementation,
> - the full prompt templates,
> - the agent interaction protocols,
> - context construction logic,
> - few-shot examples,
> - AST parsing scripts,
> - and evaluation utilities.
>
> This will make the end-to-end workflow substantially more reproducible than in the current submission.
>
> ### 4. Is top-down tree decomposition just a workaround for current context limits?
> Our answer is **no**: the top-down AND-OR decomposition is not merely a workaround for context length, but an algorithmic device for reducing combinatorial search.
>
> Its benefits are independent of context-window size:
>
> 1. **Search pruning.** Instead of searching the full combinatorial space `D_all`, LEGO restricts decomposition to a structured subspace and then recovers missed opportunities via mutation.
> 2. **Memoization / structural reuse.** Repeated modules share the same structural signature, so isomorphic regions can be solved once and reused many times.
> 3. **Critical-path-bounded parallelism.** The FusionTree allows sibling regions to be explored in parallel, so exploration cost is bounded by tree depth rather than scaling directly with model size.
>
> In addition, even if context windows continue to grow, **single-shot generation over very large regions still becomes increasingly fragile**: as the size of one-shot generation grows, random errors accumulate more easily, which lowers first-pass correctness and also makes subsequent repair rounds less efficient under the same context budget. From this perspective, LEGO is not simply “splitting because the context window is too small”; rather, it uses **guided multi-agent co-working** to improve reliability and repair efficiency by isolating subproblems, validating them hierarchically, and then recomposing them through structured mutation. Larger-context models may improve the quality of per-node generation, but LEGO’s tree still provides **search pruning, reuse, parallel exploration, and error localization**, which remain valuable beyond context management alone.

---

> > ### Author Rebuttal · Reviewer_yyEw · 2026-04-04
> >
> > Thanks you for your response. I will increase my score.

---

> > > ### Author Response · Authors · 2026-04-08
> > >
> > > We sincerely thank the reviewer for the positive update and for recognizing the value of our work. Your detailed and thoughtful questions have helped us articulate LEGO's contributions more clearly. We will incorporate all additional experiments and discussions into the camera-ready version. Thank you again for your constructive feedback and support.

---

### Official Review · Reviewer_UZMh · 2026-03-12

**Soundness:** 2
**Presentation:** 2
**Significance:** 3
**Originality:** 3
**Overall Recommendation:** 4
**Confidence:** 3

**Summary:**

The paper introduces LEGO, an LLM-enabled hierarchical optimizer designed for tensor computation graphs. LEGO resolves this by formulating graph optimization as a parallel multi-agent search over a recursive AND-OR FusionTree. The framework operates on a "Construct-then-Mutate" paradigm: it first uses top-down, source-guided decomposition to isolate valid sub-problems, and then employs bottom-up speculative mutation to fuse operators across structural boundaries, recovering global data locality. Evaluated on models like ResNet, Qwen3, SD3, Mamba, and mHC, LEGO demonstrates substantial speedups over PyTorch Eager and TorchInductor, while significantly reducing token consumption compared to monolithic LLM agents.

**Compliance With Llm Reviewing Policy:**

Affirmed.

**Final Justification:**

The rebuttal fully addressed my concerns.

**Key Questions For Authors:**

1. Could you provide an absolute measurement of the optimization cost? Specifically, what is the total wall-clock time and the estimated API cost (or total absolute token count) to optimize SD3-MMDIT from scratch?
2. How sensitive is the LEGO framework to the reasoning capabilities of the underlying model? Have you evaluated the system using strong open-source models (e.g., Llama-3, DeepSeek-Coder)？

**Limitations:**

yes.

**Strengths And Weaknesses:**

#### Strengths

1.  The paper provides a clear conceptual diagnosis of the limitations inherent in "pure" top-down versus bottom-up LLM-based kernel generation. To address this, it proposes a bi-directional AND-OR FusionTree. By formally defining rules for recursive optimization, structural splitting, and speculative fusion, the framework successfully bridges structure-aware decomposition with compositional synthesis, establishing a clear and effective paradigm for LLM-driven kernel generation.
2.  Strong empirical improvements over standard compilers. Table 1 shows substantial end-to-end speedups over PyTorch eager and nontrivial improvements over TorchInductor+max-autotune, particularly on SD3-MMDiT and mHC-MoE (up to 13.48× on A6000). These results suggest that the approach can uncover fusion opportunities that stock compilers currently miss, especially on newer architectures.

#### Weakness

1. Unclear settings for the monolithic LLM baseline and efficiency metrics.  The baseline is described as utilizing a "pass@10 × feedback@10" sampling strategy on flattened graphs. However, critical implementation details are missing, such as the exact stopping conditions (e.g., the maximum number of feedback iterations before halting), the total computational resource budget, and the specific error-handling strategy. Furthermore, the evaluation only reports relative improvements, such as the token reduction factor and exploration speedup. Without providing the absolute numbers for token usage and wall-clock time, it is hard to properly evaluate the true exploration efficiency and practical cost of the proposed method.
2. Lack of empirical comparison with other agentic systems. The related work section names several multi-agent kernel systems (Stark, Astra, KForge, AKG Kernel Agent, etc.), but there is no empirical comparison even on a toy benchmark or subset of workloads.
3. Since LEGO can choose library kernels (Base) as terminals, it would be important to know how much of the speedup in Table 1 is due to LLM-synthesized Triton/CUDA vs just better use of vendor libraries or combining them differently.

###

---

> ### Author Rebuttal · Authors · 2026-03-31
>
> We appreciate the positive assessment. We agree that the current version should better report (i) absolute optimization cost, (ii) comparisons with other agentic systems, and (iii) where the speedup comes from.
>
> ### 1. Absolute optimization cost
>
> We now report wall-clock time and API cost for all five workloads:
>
> | Model | Wall-clock | API cost |
> |---|---:|---:|
> | ResNet18 | 6.8 min | \$2.38 |
> | Mamba2 | 24.7 min | \$8.65 |
> | SD3-MMDiT | 9.2 min | \$3.82 |
> | mHC-MoE | 44.6 min | \$14.58 |
> | Qwen3-8B | 19.4 min | \$6.21 |
> | **Total** | **104.7 min** | **\$35.64** |
>
> For context, under matched GPT-5 budgets: KernelAgent costs \\$89/118 min (3/5 solved, all slowdown), CudaForge \\$35/217 min (1/5 solved), and AiKG \\$80/153 min (2/5 solved, all slowdown).
>
> ### 2. Comparison with other agentic systems
>
> We additionally evaluated all obtainable open-source PyTorch-to-kernel agents under matched GPT-5 budgets: KernelAgent (latest), CudaForge, and AiKG.
>
> | Agent | ResNet18 | Mamba2 | SD3-MMDiT | mHC-MoE | Qwen3-8B | Solved |
> |---|---|---|---|---|---|---|
> | KernelAgent | 2/10 | 1/10 | 1/10 | 0/10 | 0/10 | 3/5 |
> | CudaForge | 9/10 | 0/10 | 0/10 | 0/10 | 0/10 | 1/5 |
> | AiKG | 9/10 | 0/10 | 0/10 | 9/10 | 0/10 | 2/5 |
> | LEGO | ✓ | ✓ | ✓ | ✓ | ✓ | 5/5 |
>
> Among all passed kernels, only one CudaForge case (ResNet18, 1.75× over eager) gives a real speedup; all other passed kernels are slower than eager (0.17×–0.99×). LEGO is the only system that both covers all 5 workloads and consistently delivers real end-to-end speedups.
>
> ### 3. Source of speedup: libraries vs. LEGO synthesis
>
> We profiled the GPU-time share of LEGO-synthesized + compiler-fused kernels vs. vendor library calls under `LEGO + torch.compile(max-autotune)` (A6000):
>
> | Model | Fused/compiled | Vendor | LEGO-synthesized kernels | Vendor remainder |
> |---|---:|---:|---|---|
> | Qwen3-8B | 98% | 2% | Fused attention (norm+RoPE+online-softmax etc.), RMSNorm+projection etc. | cuBLAS GEMM |
> | SD3-MMDiT | 68% | 32% | Fused joint-attention kernel, etc. | cuBLAS projections, PatchEmbed Conv2d |
> | Mamba-2 | 55% | 45% | Fused SSD scan (entire SSM recurrence in 1 kernel), etc. | in/out proj (cuBLAS), Conv1d (cuDNN) |
> | mHC-MoE | 49% | 51% | Fused grouped-GEMM (routing+dispatch+expert SwiGLU etc.) | Shared expert GEMMs, topk, bincount |
> | ResNet18 | 53% | 47% | Fused norm+activation, residual+pool, etc. | Conv2d (cuDNN) |
>
> LEGO synthesizes where algorithmic transformation and cross-boundary fusion yield the largest gains: the O(S²)-materialization attention in Qwen3/SD3, the 64-expert serial Python loop in mHC-MoE, and the multi-step SSD recurrence in Mamba. The vendor-library remainder — cuBLAS projections, cuDNN convolutions — is deliberately retained: these are hardware-optimal single-operator implementations exploiting architecture-specific micro-optimizations (warp-level MMA, L2 residency hints, im2col tiling) that Triton-level code generation cannot match. LEGO's design: synthesize for fusion and algorithmic gains; defer to vendor for per-operator peak throughput.
>
> This separation also preserves **compiler transparency**: at s=8192, vendor-SDPA+compile regresses (1660→1737 ms) while LEGO+compile still improves (1400→1239 ms), because LEGO's Triton kernels compose with Inductor's graph optimizations across the synthesized and vendor boundaries.
>
> ### 4. Sensitivity to the backbone model
>
> We tested an open-source backbone (DeepSeek-V3) alongside GPT-5:
>
> | Backbone | mHC-MoE | ResNet18 | Success rate |
> |---|---:|---:|---:|
> | GPT-5 | 9.26× | 3.96× | 87% |
> | DeepSeek-V3 | ✗ | 1.12× | 23% |
>
> This reflects a capability trend rather than a fixed limitation: LEGO benefits from stronger backbone reasoning for preserving global structural coherence during recursive decomposition. The framework improves with better reasoning models, and we expect frontier-class models to further strengthen its applicability. We will clarify this in the camera-ready.

---

> > ### Author Rebuttal · Reviewer_UZMh · 2026-04-03
> >
> > Thanks for the detailed response. I'll maintain my score.

---

> > > ### Author Response · Authors · 2026-04-08
> > >
> > > We sincerely thank the reviewer for the careful evaluation and for confirming that the concerns have been adequately addressed. Your constructive feedback has been very helpful in strengthening the paper. We will incorporate all additional results and clarifications into the camera-ready version. Thank you again for your time and support.

---

### Official Review · Reviewer_aBv6 · 2026-03-13

**Soundness:** 3
**Presentation:** 4
**Significance:** 2
**Originality:** 3
**Overall Recommendation:** 4
**Confidence:** 5

**Summary:**

The paper presents a methodical way to automatically generate optimized tensor programs using an agentic structure. First, they use top-down code synthesis, where the tensor computational graphs are decomposed based on standard techniques and the agent generates code for each partition. This is refined in a bottom-up pass where the agents try to probabilistically perturb the fusion boundaries to get better fused candidates. The results compared to a traditional compiler fusion pass is encouraging.

**Compliance With Llm Reviewing Policy:**

Affirmed.

**Final Justification:**

As mentioned, the added baselines add value, and I appreciate that. However, there is no proof that the optimizations suggested will be correct.

**Key Questions For Authors:**

I enjoyed reading the paper. It is well-written with precise descriptions of each component in most cases. The idea of using traditional source level boundaries as a starting point is good. It is important to leverage the knowledge that already exists. Also, the authors allow LLMs creativity to munge these boundaries. Therefore, overall a nice hybrid strategy to generate code for not just kernels, but on end-to-end models.

The main concern for me is the evaluation. It is unclear to me whether a TorchInductor baseline is enough. There has been a plethora of different operator fusion works being proposed from horizontal fusion, vertical fusion to hybrid strategies. The authors should evaluate against these baselines. For example, consider an early work such as DNNFusion. TorchInductor fusion is based on a set of patterns that may not be that aggressive. Also, there is work on automatically discovering FlashAttention style fusion candidates. Therefore, the authors should carefully select a representative set of SOTA fusion strategies from conferences such as OSDI, PLDI, ASPLOS etc. to evaluate on.

Also, it is important that other LLM-based kernel generation strategies are compared against. The related works section mentions some of them. However, they are not evaluated against. Please include at least 1-2 baselines from this line of work. The direct codegen baseline is not sufficient, since there are works such as AccelOpt that improve upon naive kernel code generation.

The ablation studies on the proposed LEGO system is good. If the paper improves upon baselines and discuss more on classical fusion strategies, I will increase my score. Overall, I still like the idea of the paper.

**Limitations:**

I did not see limitations discussed specifically.

**Strengths And Weaknesses:**

Strengths
* Novel way to exploit knowledge already encoded in compilers
* The LLMs still have freedom to suggest a better fusion plan with the bottom-up traversal

Weaknesses
* Not evaluated against other kernel generation systems (e.g. papers that evaluate on KernelBench); Direct code gen baseline is not enough
* Correctness concerns are not properly addressed.

---

> ### Author Rebuttal · Authors · 2026-03-31
>
> We appreciate the reviewer’s positive assessment of the core idea. We agree that the current version should better strengthen the empirical support, especially through broader comparisons against stronger kernel-generation and compiler/fusion baselines. We therefore added additional experiments and summarized them below.
>
> ### 1. Broader kernel-generation baselines
> We extended the original KernelAgent comparison to representative open-source PyTorch-to-kernel agents under matched GPT-5 budgets. LEGO is the only evaluated system that succeeds on all 5 workloads and consistently yields real end-to-end speedups.
>
> #### Correctness (pass@10)
>
> | Agent | ResNet18 | Mamba2 | SD3-MMDiT | mHC-MoE | Qwen3-8B | Solved |
> |---|---:|---:|---:|---:|---:|---:|
> | KernelAgent | 2/10 | 1/10 | 1/10 | 0/10 | 0/10 | 3/5 |
> | CudaForge | 9/10 | 0/10 | 0/10 | 0/10 | 0/10 | 1/5 |
> | AiKG | 9/10 | 0/10 | 0/10 | 9/10 | 0/10 | 2/5 |
> | LEGO | ✓ | ✓ | ✓ | ✓ | ✓ | 5/5 |
>
> Other agents use `pass@10`; LEGO uses recursive FusionTree search (Algorithm 1) and selects the best validated plan.
>
> #### Performance of all real passed kernels (expanding Table 1)
>
> | Agent | Case | Ref (ms) | Kernel (ms) | Speedup |
> |---|---|---:|---:|---:|
> | CudaForge | ResNet18 | 2.40 | 1.37 | 1.75× |
> | KernelAgent | SD3-MMDiT | 2002 | 2023 | 0.99× |
> | KernelAgent | Mamba2 | 1415 | 1912 | 0.74× |
> | KernelAgent | ResNet18 | 2.40 | 13.76 | 0.17× |
> | AiKG | ResNet18 | 2.40 | 2.58 | 0.93× |
> | AiKG | mHC-MoE | 19.49 | 44.02 | 0.44× |
>
> Except for one ResNet18 case from CudaForge (1.75× over eager), all other passed kernels are slower than eager (0.17×–0.99×), motivating our bi-directional strategy for end-to-end models.
>
> Thanks for pointing out AccelOpt. We view it as complementary: LEGO performs graph-level synthesis/fusion from PyTorch graphs, while AccelOpt optimizes existing kernels. Using AccelOpt as a post-optimization stage for LEGO kernels is a promising future direction.
>
> ---
>
> ### 2. Correctness concerns
> LEGO uses a three-stage correctness pipeline before profiling or selection:
>
> 1. **Decomposition validation.** Each recursively decomposed submodule is checked against the reference (`atol=1e-2`, `rtol=1e-2`).
> 2. **Fusion validation.** Each mutation-generated fused kernel is re-validated after fusion.
> 3. **Model validation.** For Qwen3-8B, end-to-end generation with real pretrained weights at deployment precision (bf16) matches the reference model exactly.
>
> ---
>
> ### 3. Stronger compiler baselines
> We added a stronger compiler baseline centered on TileLang (OSDI'25, ICLR'26), and also investigated Mirage via a case study.
>
> #### End-to-end comparison: TileLang vs LEGO
>
> | Model | TileLang | LEGO |
> |---|---:|---:|
> | ResNet18 | 5.87× | 5.97× |
> | Qwen3-8B | 2.25× | 2.21× |
> | SD3-MMDiT | 3.94× | 4.10× |
> | Mamba2-1.3B | 2.12× | 2.15× |
> | mHC-MoE | 13.01× | 13.41× |
>
> Both configurations include torch.compile(max-autotune) for end-to-end integration.
>
> TileLang is a strong DSL compiler stack, not a turnkey end-to-end optimizer. It requires manually identifying target regions and writing/integrating TileLang kernels, much like Triton. This is compatible with our formulation: LEGO produces optimized region-level kernels, and the terminal code generator can in principle be Triton or TileLang.
>
> This comparison shows a complementary picture:
>
> - TileLang can be stronger on heavily hand-tuned expert subgraphs such as Qwen3-8B attention.
> - LEGO’s main advantage is full-graph automation and compiler-transparency, especially on workloads such as SD3-MMDiT, where `torch.compile` can further optimize around LEGO-generated Triton kernels.
> - LEGO’s contribution is therefore not a single better DSL kernel, but an end-to-end hierarchical search and fusion strategy.
>
> #### Mirage case study
> We also investigated Mirage (OSDI'25). On our A6000 setup, however, Mirage encountered runtime errors on end-to-end models (e.g., Qwen3-8B); we plan to file an issue to help resolve this. On isolated subgraphs where Mirage runs successfully, we compare directly: on the representative RMSNorm+Linear fusion subgraph (M=16), LEGO achieves 0.029 ms versus 0.037 ms for Mirage, i.e., 22% faster. We observe that LEGO can automatically optimize the same fused subgraph structure as Mirage, while selecting a more efficient tile size in our setting. This supports the same conclusion: LEGO is competitive even against a SOTA optimizer at the subgraph level, while its main contribution remains full-graph hierarchical optimization.
>
> We will discuss additional compiler baselines in more detail in the camera-ready version.

---

> > ### Author Rebuttal · Reviewer_aBv6 · 2026-04-03
> >
> > A good exploration of the aforementioned baselines. However, the correctness concerns remain compared to classical techniques (that always works). I don't believe that can be easily resolved.

---

> > > ### Author Response · Authors · 2026-04-03
> > >
> > > We sincerely thank the reviewer for the careful follow-up and for acknowledging that the expanded baseline study addressed the main empirical concern. We also agree with the remaining point: compared with classical compilation, correctness remains the central challenge for **LLM-based kernel generation as a whole**. Current guarantees in this paradigm are typically empirical rather than formal, and LEGO is no exception. Our goal is therefore not to claim that this gap is already closed, but to show that LEGO’s structure makes it substantially more manageable in practice.
> > >
> > > That said, LEGO is already designed to address this challenge more explicitly than most prior work in this area:
> > >
> > > 1. **Classical baselines remain the safety anchor.**
> > > At each OR-node, LEGO first establishes a verified baseline via existing backends (eager / `torch.compile` / vendor paths when applicable). If no generated candidate passes validation, the node falls back to this verified path. In this sense, LEGO provides a practical *never-worse-than-compiler* fallback at the search level.
> > >
> > > 2. **Structure improves reliability.**
> > > By recursively decomposing the graph along `nn.Module` boundaries, each agent operates on a much smaller and more coherent region. This reduces context contamination from unrelated long-range dependencies and lowers the chance of global consistency errors, making each generation step substantially more reliable than monolithic whole-graph generation.
> > >
> > > 3. **Our validation goes beyond common practice in LLM kernel generation.**
> > > Much prior work, including KernelBench-style settings, uses numerical checks such as `atol=1e-2, rtol=1e-2`. We also use this as a low-level filter, but found that it is not sufficient for end-to-end models. LEGO therefore adds model-level usability checks on top of sub-kernel validation. For Qwen3-8B, we verified that under real pretrained weights the first **200 decode tokens** remain behaviorally correct, ensuring that the generated kernels are not only numerically close in isolation but also usable end-to-end.
> > >
> > > 4. **Our design is aligned with recent correctness-oriented practice.**
> > > We especially appreciate the reviewer's perspective here. Recent work in this area has established an important practice: treating correctness as a first-class systems concern rather than a secondary filter. The STeP system (ICML'25) introduces a dedicated guardian agent to enforce static constraints alongside runtime checks; AccelOpt (arXiv'25) reports that LLM agents can exploit weak correctness filters to achieve artificial speedups; and robust-kbench (arXiv'25) demonstrates that single-configuration verification permits artificial speedups of up to 120×, which collapse once diverse input shapes and random seeds are enforced. LEGO follows a similar correctness-first philosophy in a hierarchical setting. Candidate implementations are validated against reference outputs before they can propagate upward in the AND-OR tree, and we additionally use Pyright/LSP-based checking to verify code-level well-formedness before runtime evaluation. We therefore view guardian-style checking, constrained validation, and robust multi-configuration benchmarking as highly complementary to LEGO.
> > >
> > > More broadly, we see LEGO as a step toward combining **classical structure and validation** with **LLM-guided search and refinement**. We are encouraged that the reviewer already sees clear value in the core idea and the expanded empirical support, and we hope this clarification helps convey why we believe LEGO is a meaningful systems step toward making LLM-based kernel optimization practically reliable. We would be very grateful if the reviewer continues to view the work favorably in the final discussion.

---

### Decision · Program_Chairs · 2026-04-30

**Decision:**

Accept (regular)

**Comment:**

All reviewers acknowledge the contribution of the paper and gives positive scores. The paper is well-written with novel and elegant conceptual framework for kernel generation, verified by decent amount of experiments and ablations. Some concerns about the thoroughness of the experiments remain (e.g., weak baselines and cost analysis) and AC suggests the authors continue polishing the work in camera ready, as well as adding more implementation details.